# Hypoxia-induced tracheal elasticity in vector beetle facilitates the loading of pinewood nematode

Xuan Tang[1,2†], Jiao Zhou[1†], Tuuli-Marjaana Koski[1,3†], Shiyao Liu[1,2], Lilin Zhao[1,2]*, Jianghua Sun[1,3]*

[1]State Key Laboratory of Integrated Management of Pest Insects and Rodents, Institute of Zoology, Chinese Academy of Sciences, Beijing, China; [2]University of Chinese Academy of Sciences, Beijing, China; [3]College of Life Science/Hebei Basic Science Center for Biotic Interactions, Institute of Life Science and Green Development, Hebei University, Baoding, China

**Abstract** Many pathogens rely on their insect vectors for transmission. Such pathogens are under selection to improve vector competence for their transmission by employing various tissue or cellular responses of vectors. However, whether pathogens can actively cause hypoxia in vectors and exploit hypoxia responses to promote their vector competence is still unknown. Fast dispersal of pinewood nematode (PWN), the causal agent for the destructive pine wilt disease and subsequent infection of pine trees, is characterized by the high vector competence of pine sawyer beetles (*Monochamus* spp.), and a single beetle can harbor over 200,000 PWNs in its tracheal system. Here, we demonstrate that PWN loading activates hypoxia in tracheal system of the vector beetles. Both PWN loading and hypoxia enhanced tracheal elasticity and thickened the apical extracellular matrix (aECM) of the tracheal tubes while a notable upregulated expression of a resilin-like mucin protein Muc91C was observed at the aECM layer of PWN-loaded and hypoxic tracheal tubes. RNAi knockdown of *Muc91C* reduced tracheal elasticity and aECM thickness under hypoxia conditions and thus decreasing PWN loading. Our study suggests a crucial role of hypoxia-induced developmental responses in shaping vector tolerance to the pathogen and provides clues for potential molecular targets to control pathogen dissemination.

**\*For correspondence:**
zhaoll@ioz.ac.cn (LZ);
sunjh@ioz.ac.cn (JS)

†These authors contributed equally to this work

**Competing interest:** The authors declare that no competing interests exist.

## Editor's evaluation

This valuable work explores how pathogens can cause hypoxia in insect vectors and how responses to hypoxia can be exploited to promote vector competence. Using Pine Wood Nematode (PWN) infection of pine sawyer beetles the authors demonstrate that PWN loading activates hypoxia in the vector's tracheal system. The data were collected and analyzed using solid and validated methodology.

## Introduction

Insect vectors are obligatory for the dispersal of many pathogens, including several viruses and parasites deleterious to plant and animal health. Vector competence is pivotal for the successful pathogen arrival and infection of the target host, and is determined by a sophisticated set of direct interactions as pathogens enter, spread, replicate, and retain within vector cells and tissues (***Bartholomay and Michel, 2018***; ***Gray et al., 2019***; ***Lataillade et al., 2020***). Therefore, pathogens have evolved to modulate various defensive responses of vectors, such as autophagy or apoptosis, as has been shown

**eLife digest** Various parasites, bacteria and other disease-causing pathogens are transmitted by insects. A tiny worm called the pine wood nematode, for example, is spread by pine sawyer beetles which can carry up to 280,000 worms in their trachea, the network of tubes they use to breathe. This has resulted in millions of hectares of pine forests in Asia and Europe becoming infected with the deadly disease caused by the nematodes.

Pine wood nematodes, as well as other pathogens, can exploit the biological processes of the insects carrying them to make the insects transmit them more effectively. Precisely how nematodes and other disease-causing agents do this is unclear. One possibility is that they reduce the amount of oxygen being supplied to the trachea – a phenomenon known as hypoxia – which occurs naturally at specific stages in the life of an insect, and during infections.

To test this theory, Tang, Zhou, Koski et al. used genetics and imaging approaches to study how pine wood nematodes affect the trachea of pine sawyer beetles. The experiments found that when the nematodes infected the beetles, their trachea did indeed develop hypoxia. This, in turn, made the beetles' airways more elastic and made the layer of structure lining the trachea, known as the apical extracellular matrix, thicker. These changes increased the amount of pinewood nematodes the trachea could hold, allowing the beetle to spread more worms from tree to tree.

Further experiments revealed that hypoxia in the trachea increased the levels of a protein called Muc91C in the apical extracellular matrix. When the levels of Muc91C were artificially decreased in the beetles, this made their airways less elastic and the apical extracellular matrix thinner.

This work suggests that pine wood nematodes exploit the beetles' normal responses to loss of oxygen supply to make the beetles more effective at transmitting the nematodes between pine trees. Other pathogens carried by insects may also use this strategy to help increase their transmission. Further studies on the Muc91C protein may provide clues for potential drug targets to control pine wood nematodes and protect pine trees from disease.

in some leafhoppers, whitefly, and mosquitoes, to influence vector competence (*Chen et al., 2017*; *O'Neill et al., 2015*; *Wang et al., 2020*). Although understanding the responses that pathogens induce to manipulate vector competence is important for developing efficient management strategies, the underlying mechanism remains largely elusive.

A pathogen of pine trees, *Bursaphelenchus xylophilus*, commonly known as pine wood nematode (PWN), is transmitted by cerambycid beetles of the genus *Monochamus* and causes pine wilt disease in millions of hectares of pine forests across Asia and Europe (*Akbulut and Stamps, 2012*). PWN's life cycle has two distinct developmental phases, namely the propagative stage ($L_1$-$L_4$) juveniles and reproductive adult that propagates inside pine trees and the dispersal stage ($L_{III}$ and $L_{IV}$ juveniles) characterized by its close relationship with the vector beetle that transports the nematode to new host trees (*Zhao et al., 2008*; *Zhao et al., 2014*). The dispersal $L_{IV}$ enters the tracheal system of the adult beetle through the spiracles and inhabits the trachea for several days until the vector beetle has reached a new healthy host tree. The number of dispersal $L_{IV}$ carried by an adult beetle is a critical factor influencing the damage level of the pine wilt disease in the pine host tree (*Togashi, 1985*). Extraordinarily, each primary vector beetle, such as *Monochamus alternatus* in Asia, can harbor on average 15,000 and up to 280,000 nematodes in its tracheal system (*Fielding and Evans, 1996*). In contrast to circulative pathogens that move from gut lumen to salivary glands or ovary (*Eigenbrode et al., 2018*; *Wei et al., 2017*), and to other non-circulative pathogens that typically remain in the mouthparts (*Uzest et al., 2007*) and foregut of their vectors (*Chen et al., 2011*), PWNs merely reside and do not multiply in the tracheal systems. However, the effects of PWN on its unique resident niche (i.e., the vector tracheal system), and the mechanism behind the vector's competence for loading of such large numbers of PWN are still poorly understood.

Hypoxia, which occurs when oxygen supply is inadequate, is common in mammal host tissues faced with infection by pathogens and induces several adaptive responses that improve host defense by maintaining the integrity of epithelial barriers and hindering the internalization process (*Zeitouni et al., 2016*). Recent studies suggest a modulator role of hypoxia in various biotic interactions. For example, the parasite *Trypanasoma brucei* has evolved a way to inhibit hypoxic responses to evade

the host immune response in the bloodstream (*McGettrick et al., 2016*). In contrast, a mosquito gut bacterium benefits their hosts by activating hypoxic responses improving host gut growth (*Coon et al., 2017*; *Valzania et al., 2018*). As the trachea of the adult vector beetle is packed with nematodes, reduced gas transfer ability and its consequent hypoxia are conceivable.

The insect trachea, an interconnected network of air-filled epithelial tubes characterized with optimal diameter and length, exhibits intense cycles of compression and expansion for respiration, which resembles the deflation and inflation of vertebrate lungs (*Westneat et al., 2003*). The tracheal volumes changed by this behavior increase the internal pressure for improvement of air convection, thus facilitating gas exchange and diffusion (*Hayashi and Kondo, 2018*; *Westneat et al., 2003*). To increase oxygen supply, the tracheal system adapts to hypoxia by compensatory morphological and physiological changes (*Harrison et al., 2018*), such as extensively tracheal terminal branching (*Centanin et al., 2010*) and increased compression frequency of tracheal tubes (*Greenlee et al., 2013*). Mechanical properties of the tracheal tubes are therefore prerequisites for regulating the tracheal geometry and compression capacity and mainly depend on a viscoelastic material surrounding tracheal lumen, called apical extracellular matrix (aECM) (*Dong et al., 2014*; *Hayashi and Kondo, 2018*; *Zuo et al., 2013*). In *Drosophila melanogaster*, aECMs' components, including serpentine, verm, and dumpy, affect aECM elasticity and have been reported to mediate the tubular diameter or length, possibly by restricting apical membrane growth (*Luschnig et al., 2006*; *Wilkin et al., 2000*). A study with the American cockroach (*Periplaneta americana*) suggested that chitin structures in aECM layer are responsible for the mechanical features that enable tracheal compression and volume change during respiration (*Webster et al., 2011*). Similarly, high inflation of lungs is related to the mechanical forces provided by aECM components (*Berg et al., 1997*; *Wirtz and Dobbs, 2000*). In addition, hypoxia has been shown to correlate with overall aECM composition and organization of blood vessels and respiratory networks (*Jang et al., 2021*; *Kusuma et al., 2012*), implying a possible link between hypoxia and the molecular components contributing to mechanical properties of the tracheal tubes. Taken together, we hypothesize that hypoxia may be involved in the retention of PWN in vector beetle's tracheal system through the PWN exploiting mechanical properties of the vector beetle's trachea to maintain a high vector capacity in a hypoxia-regulated manner.

Here, we revealed that the number of the nematodes in the trachea is negatively correlated with the oxygen level in the tracheal system of its vector beetle, leading to hypoxia. Both PWN loading and hypoxia enhanced the elasticity of tracheal tubes and increased the thickness of aECMs in tracheal epithelia. Based on transcriptome analysis of trachea with and without PWN, we identified aECM-related genes in the beetles and found a candidate aECM component, a resilin-like mucin protein Muc91C, the expression of which was substantially upregulated at the aECM layer after PWN loading and hypoxic treatment. *Muc91C* knockdown reduced elasticity of the tracheal tubes and aECMs thickness under hypoxic condition, indicating that *Muc91C* help regulate the aECM-related mechanical traits of the tracheal tubes. Importantly, RNAi of *Muc91C* in adult beetles significantly reduced PWN loading in trachea. These results demonstrate crucial roles of tracheal hypoxia and aECM-related tubular elasticity in improving vector competence for PWN transmission.

## Results

### PWN loading decreased oxygen levels in the tracheal system of vector beetle

To determine whether PWN loading reduces oxygen levels and caused hypoxia in trachea of vector beetles, we first measured the concentration of oxygen inside major thoracic tracheal tubes using an oxygen microsensor (*Figure 1A*). Radial oxygen profiles revealed a slight drop of oxygen pressure from 21 kPa to 18.9 kPa, respectively, when comparing pressure outside (i.e., in the ambient air) and inside the tracheal tubes of beetles without PWN. In contrast, there was a dramatic drop of oxygen pressure from 21 kPa outside of tubes to 3.8 kPa inside tubes when trachea was loaded with PWN (*Figure 1B*). Thus, our results showed that PWN loading induce oxygen shortage in trachea.

We further established the correlation between oxygen level and total number of nematodes in the trachea by measuring the oxygen concentration in the atrium cavity of the first and largest abdominal spiracles and calculating the number of nematodes in the beetle (*Figure 1A*). The partial pressure of oxygen was negatively correlated with the total number of nematodes in tracheal tissue per 0.3 g

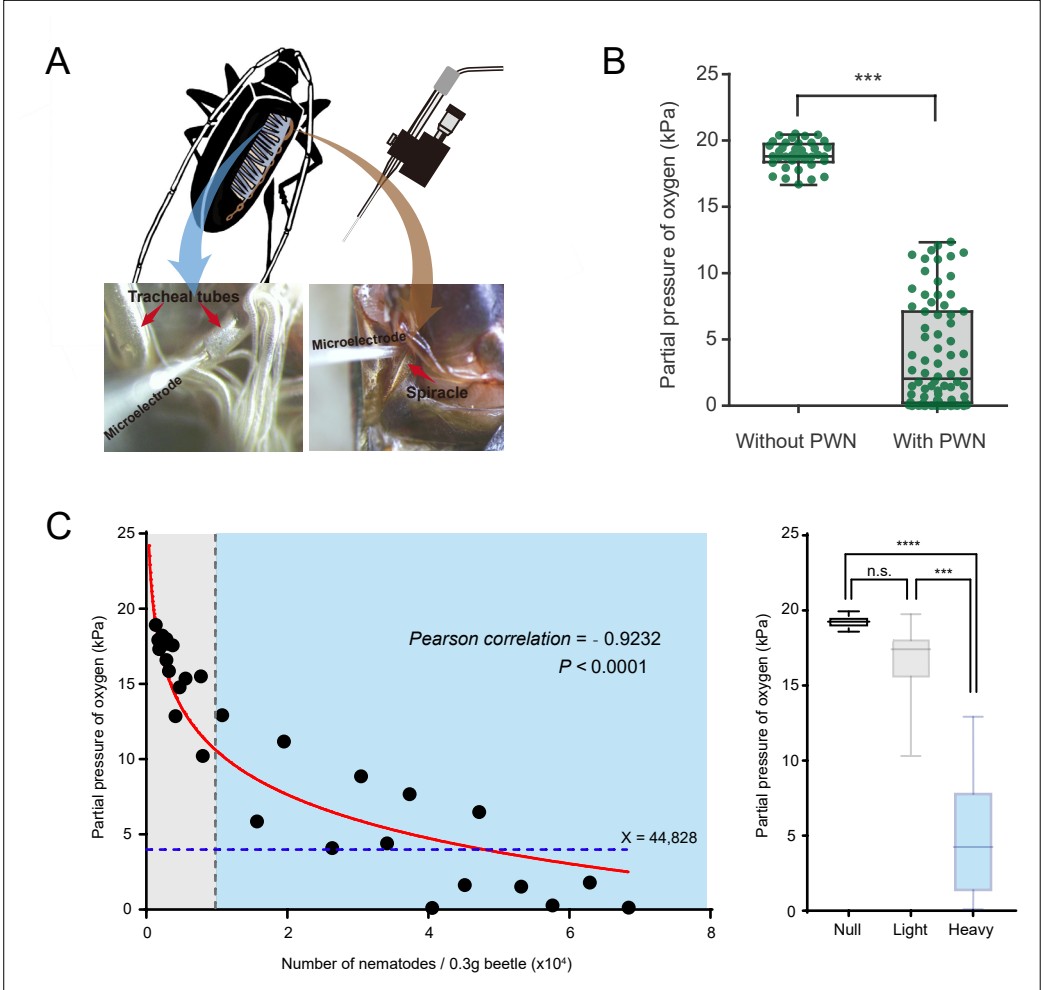

**Figure 1.** Negative correlation between oxygen level and pine wood nematode (PWN) loading in tracheal system of *M. alternatus*. (**A**) Schematic representation of experimental setup used for microelectrode (top right) measurements inside a major trachea tube of the thorax (bottom left) and the atrium cavity of the largest spiracle (bottom right). (**B**) Partial oxygen pressure in the tracheal tubes of vector beetles with and without PWN (N = 38 and 76 tracheal tubes in five adults, respectively). ***p<0.0001 (Mann–Whitney U test). The internal lines are medians, edges of the boxes are interquartile ranges, and whiskers are minimum and maximum ranges. (**C**) The relationship between oxygen partial pressure in atrium of vector beetle (N = 5, 17 and 14 adults for null, light, and heavy PWN loading) and the number of nematodes. Left panel: fitted curve is drawn in red line. The gray line, which shows 10,000 PWN, divides beetle samples into light or heavy PWN loading. The blue line shows the estimated loading number of nematodes when partial pressure drops to 4 kPa. Right panel: partial pressure of oxygen in atrium of beetles in three different degrees of PWN loading. ***p=0.0001, ****p<0.0001 (Kruskal–Wallis nonparametric test, Dunn's multiple comparison test). The internal lines are medians, edges of the boxes are interquartile ranges, and whiskers are minimum and maximum ranges.

The online version of this article includes the following source data for figure 1:

**Source data 1.** Raw data for partial pressure of oxygen and corresponding PWN number.

beetle (*Figure 1C*). The curve fit to this data revealed that when the total number of nematodes reached 10,000, partial pressure of oxygen in the cavity of tracheal spiracle dropped to 10 kPa (half of the corresponding value in the surrounding air usually 21 kPa), creating a mild hypoxia at approximately $PO_2$ of 5–15 kPa. In addition, when the number of nematodes increased to 44,828, the oxygen pressure dropped dramatically to 4 kPa (*Figure 1C*), which is the threshold between mild and severe functional hypoxia in *Drosophila* (*Harrison et al., 2018*). Oxygen levels in the cavity of the atrium were further compared among three levels of PWN loading, differing in total number of nematodes in the trachea. The tracheal systems of beetles without PWN (null) or with light PWN loading (<10,000)

had an oxygen pressure equal to that of the environment. By contrast, substantial decline to approximately 4 kPa of oxygen were detected in the tracheal spiracle with heavy PWN loading (>10,000). These results indicated PWN loading caused oxygen loss and light or heavy PWN loading induced hypoxia or anoxia in the trachea of vector beetles.

## Nematode-induced hypoxia enhanced tracheal elasticity and thickened its apical extracellular matrix

Compared to those free of PWN, the trachea of vector beetles bearing PWN tended to be rubberlike (*Figure 2—figure supplement 1A*). The total number of the nematodes in the trachea correlated positively with the tracheal rubberization degree (*Figure 2—figure supplement 1B*). Trachea with the highest rubberization degree released five times more nematodes (reaching approximately 16,000 nematodes on average) compared to trachea with moderate or low rubberization. Preliminary observations further showed that the trachea tubes harboring PWN tended to be longer and narrower before breakage under external force (*Figure 2—figure supplement 2*, *Figure 2—video 1*, and *Figure 2—video 2*). This relationship indicated that PWN loading influenced the mechanical properties of the vector's trachea.

The elasticity of trachea tubes was further quantified by comparing Young's modulus values of tracheal tubes among the three levels of PWN loading (null, light, and heavy) mentioned above. Corresponding to their higher oxygen level, Young's modulus values of trachea tubes without PWN and with light PWN loading were almost two times higher than the values in hypoxic trachea tubes with heavy PWN loading (*Figure 2A*). Given the negative correlation between elasticity and Young's modulus values, this result suggested that heavy PWN loading enhanced tracheal elasticity. To confirm the influence of hypoxia on tracheal elasticity, we subjected beetles free of PWN to 1% $O_2$ for 6 hr, 12 hr, and 24 hr. Compared with that of tracheal tubes exposed to normoxia and hypoxia conditions for 6 hr, the Young's modulus values under hypoxic treatments for 12 hr and 24 hr decreased by nearly a half (*Figure 2A*). These results clearly demonstrated that heavy PWN loading and its induced hypoxia enhanced tracheal elasticity.

Due to the correlation between mechanical properties and aECM, we further investigated the tracheal structure with null and heavy PWN loading using transmission electron microscopy. Micrographs revealed that the secreted protein layer of tracheal aECM was significantly thickened by heavy PWN loading (*Figure 2B and C*). However, the heavy PWN loading had no effect on the morphology of the chitin layer of aECM. To examine the effect of hypoxia on tracheal aECM, we treated beetles free of PWN with 1% $O_2$ for 6 hr, 12 hr, and 24 hr. Electron micrographs showed that the secreted protein layer of tracheal aECM under hypoxia for 12 hr and 24 hr was thicker than those under normoxia and hypoxia for 6 hr (*Figure 2C*). Therefore, our results indicated that PWN-induced hypoxia promoted the secretion of protein layers in aECM in trachea. Taken together, these results suggest that the increased secretion of protein layer of aECM was involved in enhanced tracheal elasticity by PWN-induced hypoxia.

## PWN loading caused significant upregulation of Muc91C responsible for the thickened aECM layer in trachea

To confirm the role of proteins to the enhanced tracheal elasticity, we compared the tracheal transcriptomes between PWN-free and heavily PWN-loaded beetles and discovered 916 differentially expressed genes (DEGs). Out of these, 251 DEGs were identified as aECM-related genes (*Figure 3A*). Overall, the altered transcript profiles suggested dramatic epithelium reorganization in tracheal tubes of vector beetles after PWN loading. We then chose 45 genes related to mechanical properties to further scrutinize for their expression (*Figure 3B*). We examined chitinases, chitin synthases, and chitin deacetylases, which are involved in chitin catabolic processes, biosynthetic processes, and arrangement, respectively (*Dong and Hayashi, 2015*). The up- or downregulated pattern in the expression of these transcripts suggested a chitin metabolic equilibrium, which was consistent with the unchanged chitin layer among beetles with heavy PWN loading shown in TEMs (*Figure 2C*). Our gene list also included those genes functioning as aECM regulators. For example, metalloproteinases are proteinases that cleave proteins in the extracellular matrix (*Glasheen et al., 2010*). Cadherins, integrins, lachesin, and dumpy are transmembrane proteins participating in signal transduction between aECM and tracheal cells (*Llimargas et al., 2004*; *Öztürk-Çolak et al., 2016*; *Wilkin et al., 2000*).

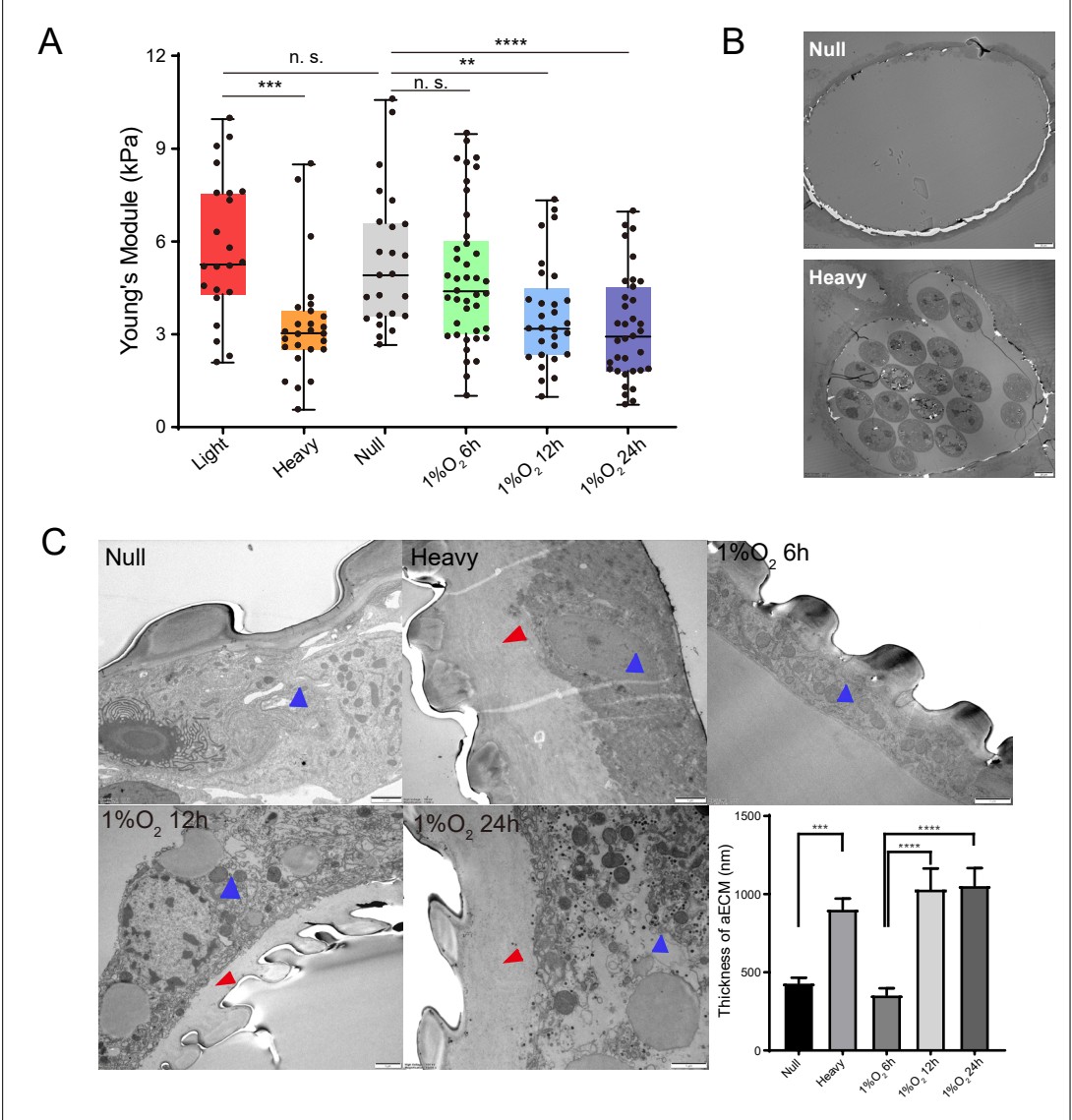

**Figure 2.** Effects of pine wood nematode (PWN) loading and hypoxia on mechanical properties and apical extracellular matrix (aECM) of the trachea tubes of *M. alternatus*. (**A**) Calculated Young's modulus in different experimental groups according to values in tensile test in the longitudinal direction on the longest tubes linking two homolateral spiracles. Groups represent tubes from beetles without PWN (Null), light (<10,000 nematodes), heavy (>10,000 nematodes) PWN loading, and treated with 1% $O_2$ for 6 hr, 12 hr, and 24 hr, respectively (N = 22–41 tracheal tubes for each treatment). **p<0.01, ***p<0.001, and ****p<0.0001 (Kruskal–Wallis nonparametric test, Dunn's multiple comparison test). Internal lines are medians, edges of the boxes are interquartile ranges, and whiskers are minimum and maximum ranges. (**B, C**) Transmission electron micrographs (TEMs) for axial views of the longest tracheal tubes. (**B**) Representative images were shown for global view of tracheal tubes without (null) and with heavy PWN loading. Bars, 20 μm. Experiments were performed at least three times. (**C**) Electron microscopy images showed the thickened tracheal aECM (red arrow) between the chitin layer (taenidial ridges) and the body of tracheal cell (blue arrow) after different treatments (without PWN, i.e., null, heavy PWN, and 1% $O_2$ for 6 hr, 12 hr, and 24 hr, respectively). Bars, 1 μm. The right bottom panel shows the statistical data of the thickness of aECM corresponding to TEMs. Data were measured in six tracheal tubes of three adults for each treatment. ***p<0.001 and ****p<0.0001 (Kruskal–Wallis nonparametric test, Dunn's multiple comparison test). Data are represented as mean ± SEM.

The online version of this article includes the following video, source data, and figure supplement(s) for figure 2:

**Source data 1.** Raw data for Young's modulus and aECM thinkness in different experimental groups.

**Figure supplement 1.** Morphological changes of trachea of *M. alternatus* after pine wood nematode (PWN) loading.

**Figure supplement 1—source data 1.** Raw data for the relationship between PWN number and rubberization degree.

**Figure supplement 2.** Pulling test of trachea of *M. alternatus* after pine wood nematode (PWN) loading.

**Figure 2—video 1.** Pulling test of trachea of *M. alternatus* without pine wood nematode (PWN).

*Figure 2 continued on next page*

**Figure 2—video 2.** Pulling test of trachea of *M. alternatus* with pine wood nematode (PWN).

The changed expression of these transcripts after heavy PWN loading indicated altered regulatory cascades of aECM. We finally investigated the expression of the non-chitin structural components, including mucins, collagens, and laminin. Negatively regulated transcripts of collagens in heavy loading PWN samples and the location of laminin in basal lamina of tracheal cells (*Dai et al., 2018*) suggest they were not responsible for thickening the aECM layer in our observations (*Figure 2C*). Therefore, the genes encoding for the mucin family, a group of large glycosylated macromolecules, are possible candidate genes contributing to the elasticity of trachea in our study because their location and upregulation directly corresponded to the observed thickened non-chitin layer in aECMs visible in electron microscopy (*Figure 2C*). To investigate possible correlation between tracheal development at metamorphosis and after PWN loading, we further scrutinized the expression of the 45 aECM genes in tracheal tubes after 1 day, 3 days, 5 days, and 7 days post eclosion using RNA-seq data in a published

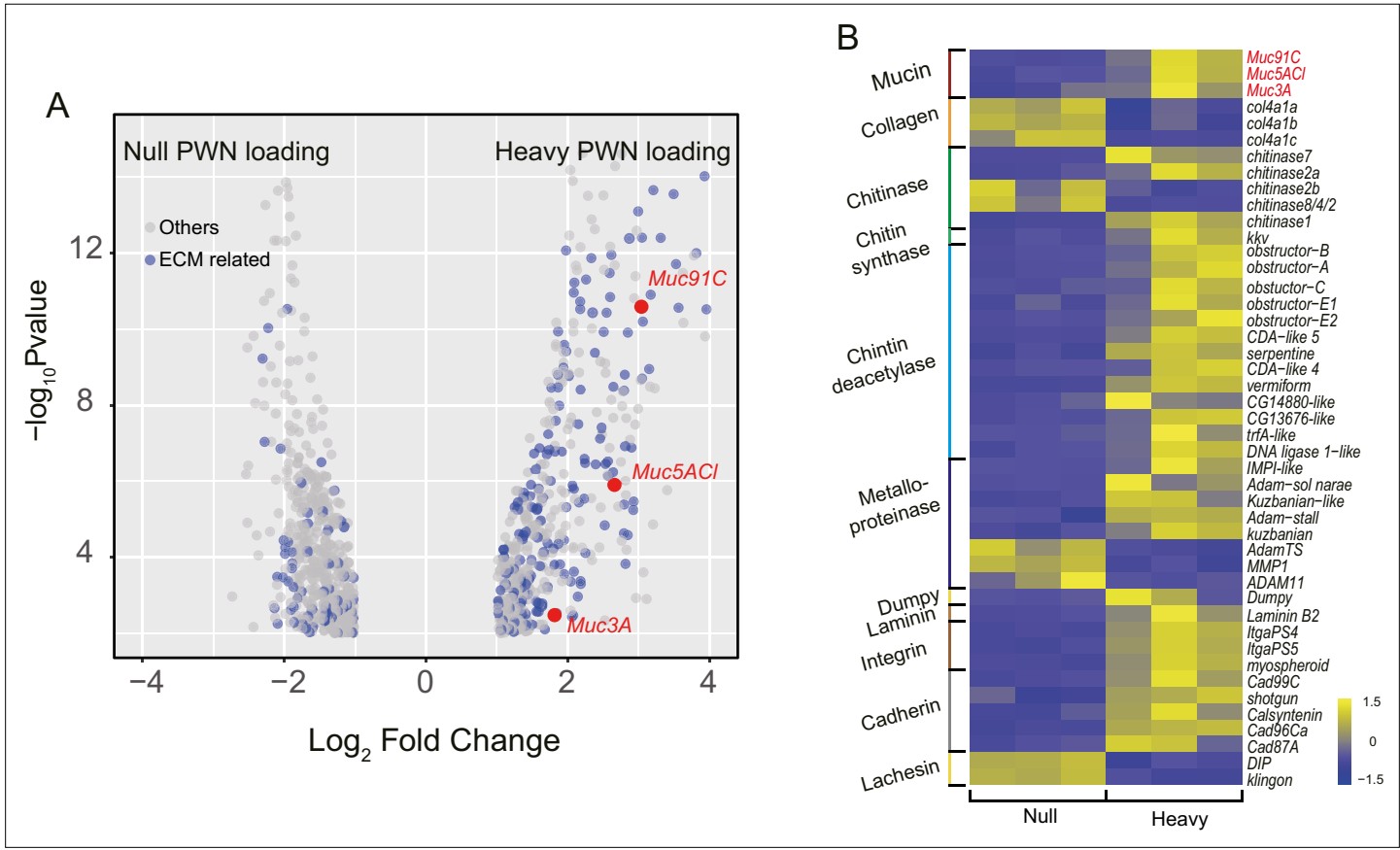

**Figure 3.** Transcriptome analysis showed changed expression pattern of apical extracellular matrix (aECMs)-related genes in trachea treat with pine wood nematode (PWN). (**A**) Volcano plot of RNA-seq data from trachea of beetles with and without PWN. Gray dots indicate all differentially expressed genes (DEGs), and blue dots indicate aECM-related DEGs. The red dots indicate the three mucins. (**B**) Heat map of differential expressed genes related to mechanical property of tracheal tubes of beetles with and without PWN. The red font indicates the three mucins.

The online version of this article includes the following source data and figure supplement(s) for figure 3:

**Source data 1.** Raw data for FPKM values of 45 selected aECM genes in trachea with and without PWN.

**Figure supplement 1.** Heat map of differential expressed genes related to mechanical property of tracheal tubes of beetles after 1 day, 3 days, 5 days, and 7 days post eclosion (PAE 1, PAE 3, PAE 5 and PAE 7, respectively).

**Figure supplement 1—source data 1.** Raw data for FPKM values of 45 selected aECM genes during tracheal maturation post adult eclosion.

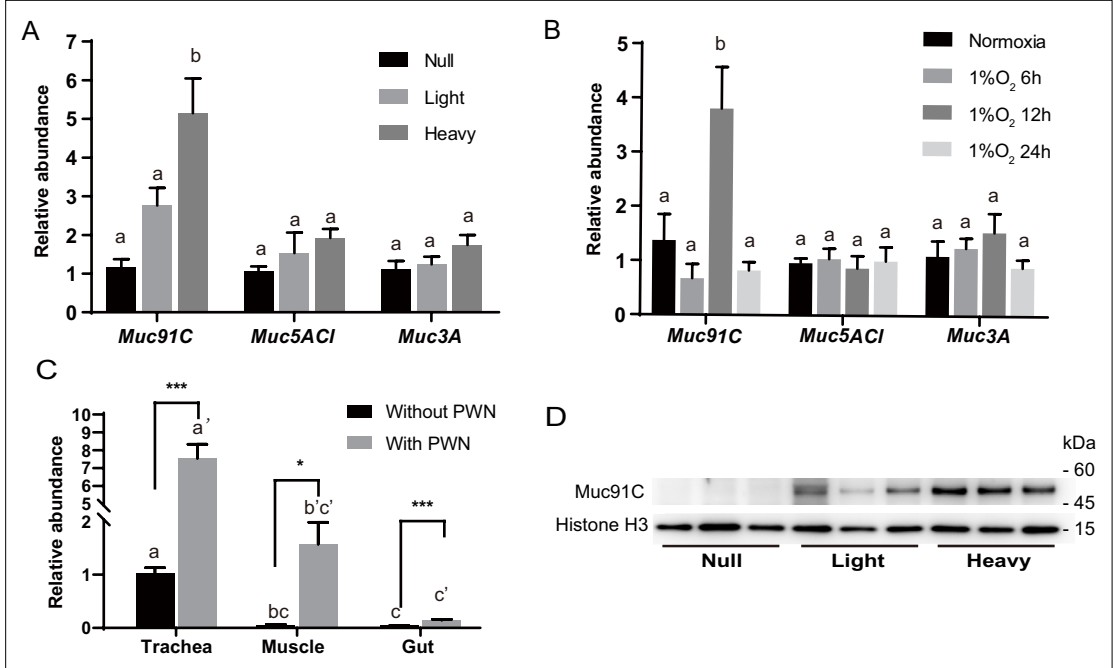

**Figure 4.** Heavy pine wood nematode (PWN) loading and hypoxia enhance Muc91C expression at the mRNA and protein level. (**A**) Relative RNA abundance of three genes belonging to mucin family in trachea of beetles with null, light, (<10,000 nematodes) or heavy (>10,000 nematodes) PWN loading. Data are represented as mean ± SEM. Columns labeled with different letters indicate statistically significant differences in mean relative abundance (N = 7–8 for each gene, one-way ANOVA with Tukey's multiple comparisons, p=0.0003, 0.1992, and 0.1353 for *Muc91C*, *Muc5ACl*, and *Muc3A*, respectively). (**B**) Relative RNA abundance of three genes belonging to mucin family in trachea of beetles under normoxia or 1% $O_2$ for 6 hr, 12 hr, or 24 hr. Data are represented as mean ± SEM. Columns labeled with different letters indicate statistically significant differences in mean relative abundance (N = 5 for each gene, one-way ANOVA with Tukey's multiple comparisons, p=0.0009, 0.9389, and 0.3662 for *Muc91C*, *Muc5ACl*, and *Muc3A*, respectively). (**C**) Relative RNA abundance of *Muc91C* in tracheal tubes, flight muscle, and midgut of beetles with null or with heavy PWN loading. Data are represented as mean ± SEM. *p<0.05, ***p<0.001 (N = 4, Mann–Whitney nonparametric test). Columns labeled with different letters indicate statistically significant differences in mean relative abundance (N = 4, one-way ANOVA with Tukey's multiple comparisons, p<0.0001 for beetles with or without PWN). (**D**) The protein level of Muc91C in tracheal tube with null, light, and heavy PWN. Histone H3 is used as internal control. Three biological replicates of each treatment are shown.

The online version of this article includes the following source data for figure 4:

**Source data 1.** Raw data for mRNA level of *Muc91C* in different experimental groups and tissues.

**Source data 2.** Raw data for protein level of Muc91C in tracheal tubes with PWN.

study (*Tang et al., 2022*). All genes were downregulated between day 5 and day 7 post adult eclosion (*Figure 3—figure supplement 1*). Therefore, tracheal system of vector beetles responded to loading nematodes despite the completion of tracheal development. For *Muc91C* specifically, its expression was twofold upregulated 3 days after eclosion but was downregulated during the later stages post eclosion. However, PWN loading resulted in a sevenfold increase of *Muc91C* expression, a higher fold change than that during tracheal development.

Among the three mucin genes, *Muc91C* had the highest increase in expression after PWN loading, increasing more than sevenfolds (log2FC = 2.43), higher than the other two genes (*Figure 3A*). Next, we independently validated our RNA-seq results by repeating the experiment using new individuals and quantifying expression using reverse-transcription qPCR (RT-qPCR) for the genes that belong to the mucin family. In contrast to the other two mucin genes, the abundance of *Muc91C* transcripts positively correlated with the level of PWN loading in trachea (*Figure 4A*). Compared to that of trachea without PWN, the expression level of the *Muc91C* gene increased fivefold in trachea with heavy PWN loading. To test the effect of hypoxia on *Muc91C*, *Muc5ACl*, and *Muc3A*, gene expression levels were measured after exposure of beetles under 1% $O_2$ for 6 hr, 12 hr, and 24 hr. Hypoxia treatment of 12 hr caused significant upregulation of *Muc91C* in trachea (*Figure 4B*). To exclude the influence of related tissues, we examined the tissue specificity of *Muc91C* in tracheal tubes, flight muscle, and midgut. Compared to fight muscle and midgut, *Muc91C* expression was mostly highly expressed in

the trachea of adult beetles with or without PWN loading. Heavy PWN loading significantly increased *Muc91C* expression in all tested organs (*Figure 4C*), implying systematic hypoxia in beetles with heavy loading. We also further confirmed the protein levels of *Muc91C* in tracheal tubes with null, light, and heavy PWN loading by Western blot analysis. The PWN loading enhanced Muc91C in a nematode number-dependent manner (*Figure 4D*), similar with the tendency of oxygen loss in tubes with PWN

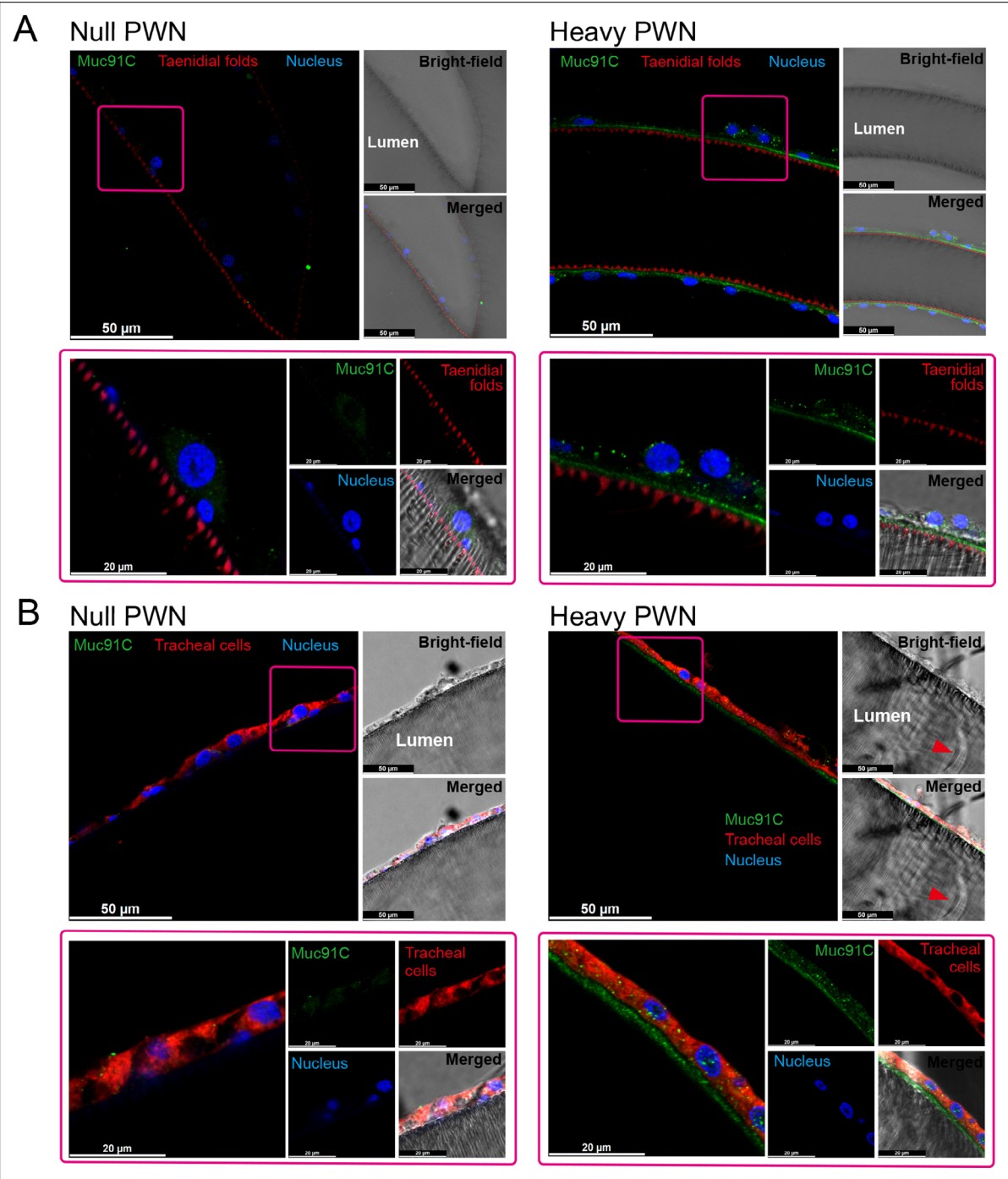

**Figure 5.** The localization of Muc91C at the apical extracellular matrix (aECM) layer in tracheal tubes. Immunostaining images of tracheal tube with null (left panels) and heavy pine wood nematode (PWN) (right panels) loading. Boxed regions (pink) in the top-left fluorescence images (bars, 50 µm) are shown at high magnification at below panels (bars, 20 µm), illustrating the localization of Muc91C (green), taenidial folds (red in **A**), tracheal cell body (red in **B**), and nuclei (blue). Bright-field images and merged bright-field and fluorescent images are shown at the right of each panel. The red arrowheads in bright-field image show residual PWN in the tracheal tube. Experiments were performed at least three times.

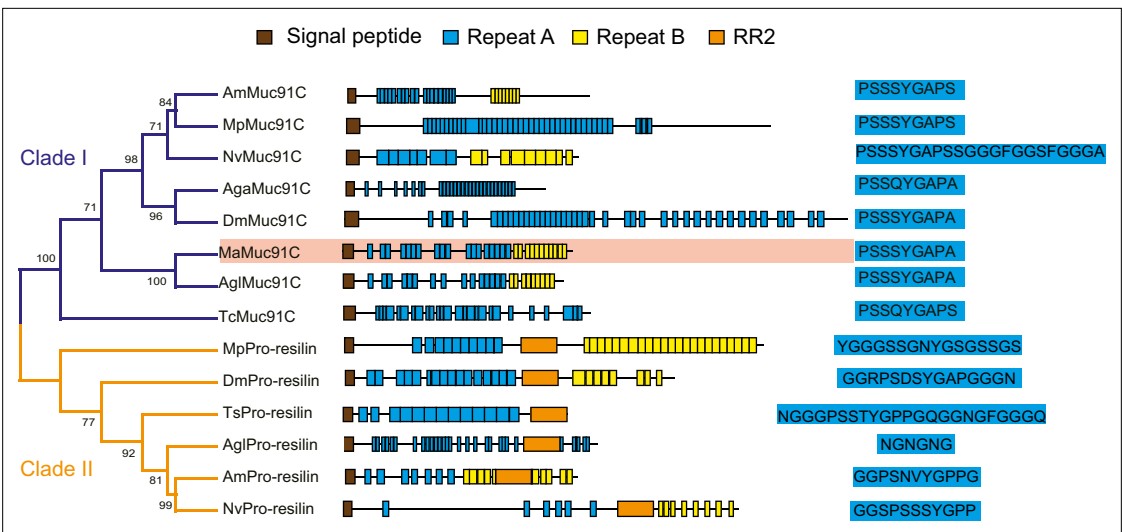

**Figure 6.** Phylogenetic relationship among resilin-related proteins in insects. The phylogenetic analysis was performed using the full-length amino acid sequences of representative resilin-related proteins. Am, *Apis melifera*; Mp, *Myzus persicae*; Nv, *Nasonia vitripennis*; Aga, *Anopheles gambiae*; Dm, *Drosophila melanogaster*; Ma, *Monochamus alternatus*; Agl, *Anoplophora glabripennis*; TC, *Tribolium castaneum*. The blue and yellow branches represent Muc91C (Clade I) and Pro-resilin (Clade II), respectively. The domain architectures are listed at the middle panel. Brown boxes indicate signal peptide. Blue boxes indicate repeats A. Yellow boxes indicate repeats B. Orange boxes indicate RR2. The consensus sequences of repeats A are listed at the right panel. MaMuc91C is highlighted in pink.

The online version of this article includes the following source data and figure supplement(s) for figure 6:

**Source data 1.** Raw data for sequences for MaMucins.

**Figure supplement 1.** A phylogenetic tree constructed using *M. alternatus* mucin sequences (orange dots) and other species' mucin sequences.

(*Figure 2C*). These results indicated that heavy PWN loading in beetles' trachea stimulates the expression of Muc91C via inducing hypoxia.

We further investigated the localization of Muc91C in tracheal tubes through immunohistochemistry analysis. Strong signals for Muc91C were detected beneath the layer of taenidial folds (*Figure 5A*) and above the apical membrane of the tracheal cell (*Figure 5B*) forming a continuously layer in tracheal tubes with heavy PWN, whereas slight fluorescence signal was observed in tracheal tubes without PWN. Thus, this result demonstrated that Muc91C is the component of the PWN-thickened aECM layer showed in TEM.

## *Muc91C* contributes to hypoxia-enhanced trachea elasticity and promote PWN loading

In order to determine whether increased Muc91C contribute to the tracheal elasticity, we investigate its gene structure and revealed that *Muc91C* encoded a resilin-like protein (*Andersen, 2010*), dominated by two long regions containing a series of short repeat motifs (*Figure 6*). Seventeen A repeats, with the consensus sequence PSSSYGAPA(S), were located in the N-terminal region. Additionally, 11 B repeats, with the consensus sequence GGYSSGGN, were located in the C-terminal region. As the only identified MaMuc91C homolog in insects, DmMuc91C was classified as a resilin-like protein (*Andersen, 2010*). In addition, a BLASTX search of MaMuc91C against Flybase revealed its similarity not only to DmMuc91C (CG7709-PB) but also to functional DmPro-resilin (CG15920-PA). Using DmPro-resilin and resilin-like DmMuc91C as query sequences in turn, we further characterized 17 resilin-related genes in the tracheal transcriptome of the PWN vector beetle and in the genomes of other insects. Phylogenetic analysis revealed that MaMuc91C and six proteins from other insects were assigned to clade I, which share repeat A motif with a consensus sequence of PS(Q)SSYGAP(A)S. Importantly, repeat A motifs in clade I shared two similar features with pro-resilin proteins. First, the glycine and proline residues included in the repeat A motifs in this clade potentially exhibit long-range elasticity by forming a stretchable beta-spiral structure (*Ardell and Andersen, 2001*; *Tatham and Shewry, 2002*). Secondly, tyrosine residues also included in repeat A motifs in clade I presumably

facilitates the formation of dityrosine crosslinks with other repeats after secretion from the epidermal cells (*Andersen, 2010*; *Qin et al., 2009*). However, clade I lacked the chitin binding R&R Consensus sequence, a defining feature of true pro-resilin. Thus, these proteins were classified as resilin-like proteins. Based on the PTS domain and O-glycosylation sites (*Zhao et al., 2020*), we found that all the proteins in clade I (*Figure 6*) belonged to the mucin family and may be homologues of Muc91C, given their similarity with DmMuc91C. The other six resilin-related proteins were true pro-resilins and clustered into clade II, in which repeat A motifs were highly variable and all sequences contained the chitin binding R&R Consensus. No pro-resilin homologue was identified in tracheal transcriptome of *M. alterantus*. Regardless of the existence of R&R Consensus sequence, repeat A consensus in both pro-reslins and Muc91C likely confer long-range elasticity because previous studies have shown that synthetic peptide chains consisting of either 17 copies of A repeats (GGRPSDSYGAPGGGN) from DmPro-resilin or 16 copies of A repeats (AQTPSSQYGAP) from the AgaMuc91C (AGAP002367-PA), were able to form rubberlike elastic materials (*Elvin et al., 2005*; *Lyons et al., 2007*; *Nairn et al., 2008*). In addition, phylogenetic analysis based on *M. alternatus* and other insect species' mucin sequences showed that Muc91C proteins were clustered into a separate lineage (*Figure 6—figure supplement 1*). We therefore deduced that *MaMuc91C* encodes an insect-originated resilin-like protein and has a potential role in providing long-range elasticity.

We further verified the key role of *Muc91C* in determining the thickness of aECM and its related elasticity under hypoxia condition. Firstly, we used TEM to observe the substructure of tracheal tubes treated with 1% $O_2$ for 12 hr after performing gene knockdown of *Muc91C* in adult beetles. RNAi knockdown of *Muc91C* in adult beetles within 2 days after eclosion resulted in a 77.5% decrease in *Muc91C* mRNA levels and has no detrimental influence on adult survival (*Figure 7—figure supplement 1A and B*). Compared to ds*GFP*-injected beetles, the thickness of aECM layers in tracheal tubes were substantially reduced in ds*Muc91C*-injected beetles (*Figure 7A*). Next, we measured Young's modulus values of trachea tubes treated with 1% $O_2$ for 12 hr after dsRNA injection. Compared to the ds*GFP* control, the Young's modulus values of tracheal tubes were significantly increased in the ds*Muc91C* samples, indicating their decreased elasticity after *Muc91C* knockdown (*Figure 7B*). Therefore, the enhanced tracheal elasticity induced by hypoxia relies on the expression of *Muc91C*.

To further confirm the role of *Muc91C*-induced elasticity in PWN loading, we carried out *Muc91C* knockdown in adult beetles and subsequently inoculated beetles with PWN. Injection of *dsGFP* and ds*Muc91C* resulted in similar proportions of death before dissection for PWN counting (*Figure 7—figure supplement 1C*). Compared to beetles injected with ds*GFP*, the number of nematodes in beetles injected with ds*Muc91C* was reduced by 50%, and the total number of nematodes in these beetles' trachea reaching less than 10,000 (*Figure 7C*). Collectively, *Muc91C* was essential to promote heavy PWN loading by promoting tracheal elasticity during hypoxia.

## Discussion

Vector competence is important for vector-borne pathogens' transmission, but the underlying mechanism remains largely unexplored. Here, we demonstrate one of the strategies that PWN utilizes to maintain itself in its vector by modulating the vector beetle's hypoxia responses. This strategy helps the nematode to reach high numbers even in a single vector beetle and may consequently promote the successful spread, establishment, and outbreak of pine wilt disease. Mechanically, we propose that PWN's loading process in vector beetles follows a positive feedback loop (*Figure 8*). First, continuous entering of nematodes into trachea decreases oxygen levels, resulting in hypoxia, which in turn increases the elasticity of tracheal tubes by upregulating *Muc91C*. The more elastic tube structure, in turn, allows more nematodes to reside in the lumen until maximum elasticity is reached, resulting in decreased gas transfer due to physical blocking caused by nematodes. Finally, the resulting high level of $CO_2$ drives nematodes away from the trachea (*Wu et al., 2019*). Thus, the nematodes strategically manage their loading and departure from the vector by manipulating oxygen supply in the vectors' trachea.

Hypoxia occurs universally across living organisms and plays multiple roles in hosts infected by pathogens. For instance, a protozoan parasite *Leishmania donovani* induces inflammatory hypoxia to limit the capacity of the host's myeloid cells to kill the parasite (*Hammami et al., 2017*). Some commensal gut microbiota maintains epithelial hypoxia to limit the growth of enteric pathogens in the gut lumen of neonatal chicks (*Litvak et al., 2019*). However, hypoxia in pathogen–vector relationships

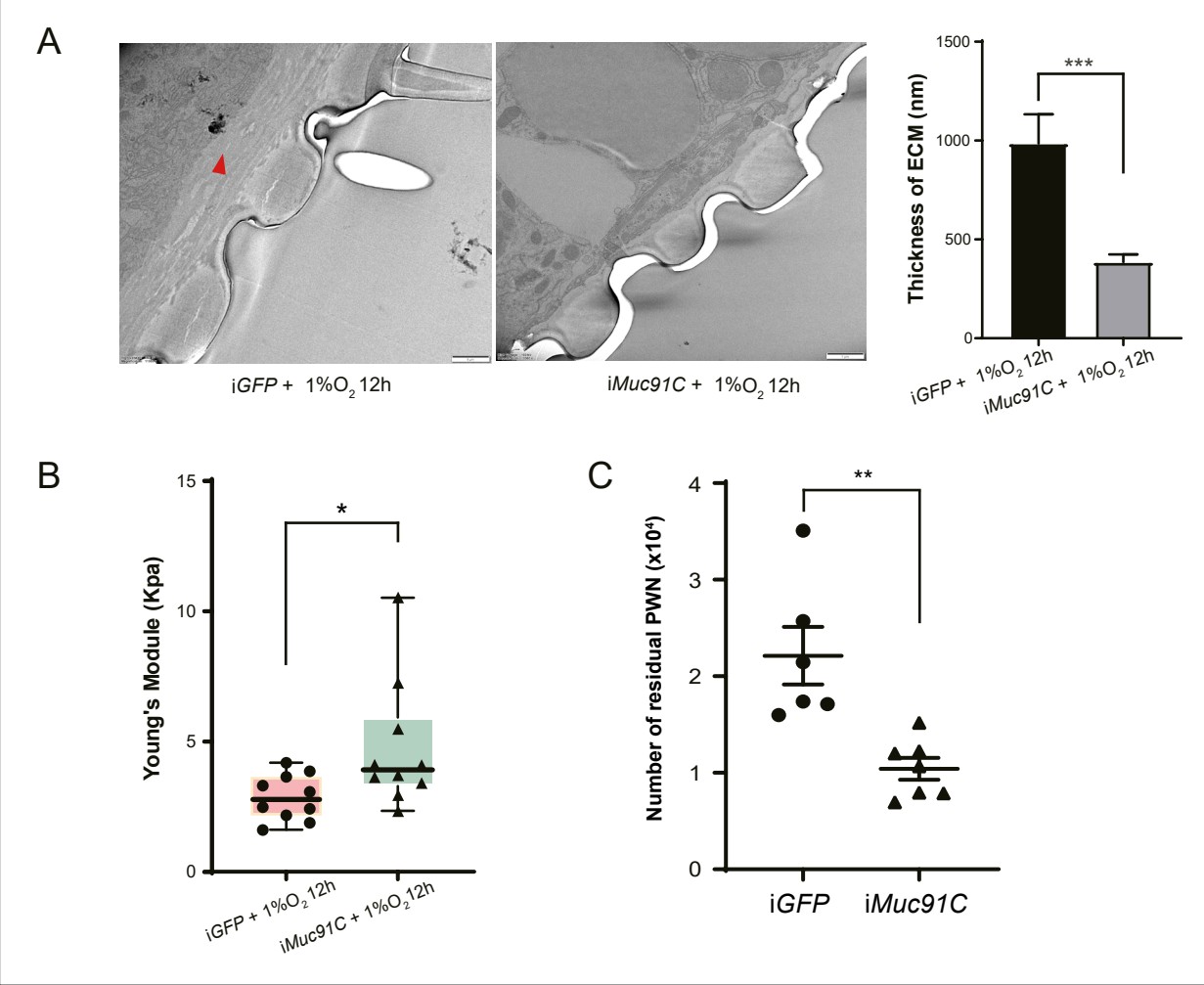

**Figure 7.** The role of *Muc91C* in tracheal elasticity and pine wood nematode (PWN) loading. (**A**) Electron microscopy images showed the thickened tracheal apical extracellular matrix (aECM) (red arrow) under 1% $O_2$ for 12 hr after ds*GFP* or ds*Muc91C* injection. Bars, 1 µm. The right panel shows the statistical data of the thickness of aECM corresponding to TEMs. Data were measured in six tracheal tubes of three adults for each treatment. ***p<0.001 (Mann–Whitney nonparametric test). Data are represented as mean ± SEM. (**B**) Young's modulus of trachea treated 1% $O_2$ for 12 hr after ds*GFP* or ds*Muc91C* injection (N = 10 tracheal tubes for each treatment). *p<0.05 (Mann–Whitney nonparametric test). The internal lines are medians, edges of the boxes are interquartile ranges, and whiskers are ranges. (**C**) Number of residual PWNs released from every 0.3 g beetle incubated with PWN (N = 6 and 7 adults, respectively). ds*GFP* or ds*Muc91c* is injected before incubation. **p<0.01 (Student's *t*-test, unpaired and two-tailed). Data are represented as mean ± SEM.

The online version of this article includes the following source data and figure supplement(s) for figure 7:

**Source data 1.** Raw data for aECM thickness, Young's Module values and PWN numbers of ds*Muc91C*-treated beetles.

**Figure supplement 1.** RNAi efficiency of the Muc91C gene and survival rates in dsRNA-treated adult beetles.

**Figure supplement 1—source data 1.** Raw data for mRNA level of *Muc91C* and survival rate of ds*Muc91C*-injected beetle.

has long been a neglected research topic. Our study shows that retention of nematodes caused hypoxia in tracheal tubes of vector beetles, providing empirical evidence for a previous hypothesis that the heavy PWN loading hinders the gas exchange of the beetle (*Togashi and Sekizuka, 1982*). Such hypoxia induced by PWN loading improved the vector's tracheal capacity to harbor PWN. The correlation between hypoxia and vector competence can be extrapolated to the interaction between circulative pathogens and their vectors because their migration, replication, or propagation processes inside vectors may also increase oxygen demands and result in hypoxia.

Mechanical properties of tracheal tube are critical to luminal volume by regulating tubular size (*Dong et al., 2014*) and the dynamic volume changes during tracheal compression and expansion to

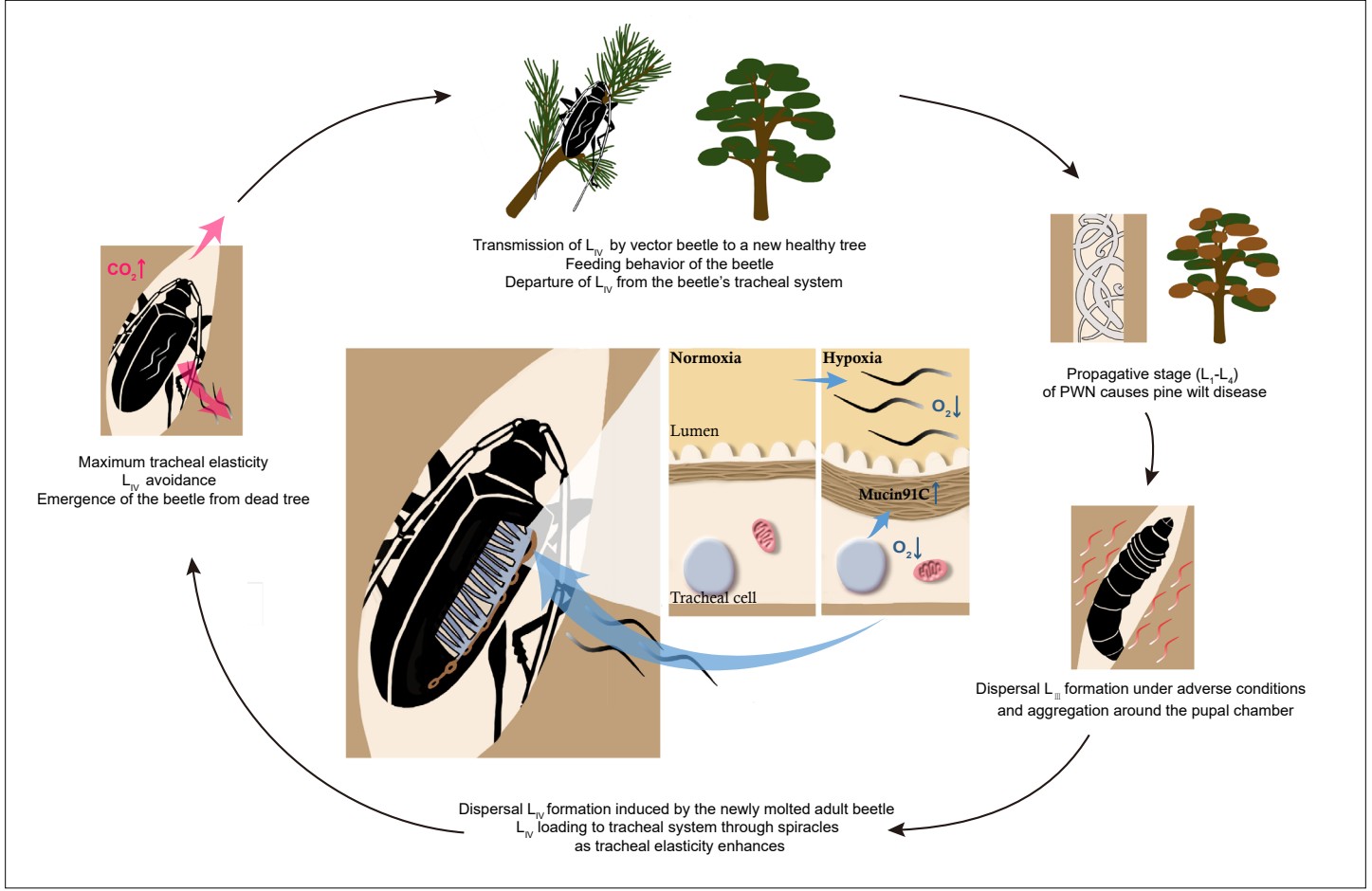

**Figure 8.** Schematic diagram of the feedback regulation of pine wood nematode (PWN) loading through $O_2$ and $CO_2$ level in the tracheal system of *M. alternatus*. When $L_{IV}$ dispersal nematodes continuously enter the vector beetle through spiracles, the consequent hypoxia enhances the elasticity of tracheal tubes by upregulating *Muc91C*. The more elastic tube structure, in turn, allows more nematodes to reside in the lumen. When tracheal tubes reach to their maximum elasticity, nematodes influence gas exchange of beetles and the resultant high level of $CO_2$ drives nematodes away from the trachea.

increase the internal pressure for air convection and gases exchange, thus facilitating oxygen diffusion (*Westneat et al., 2003*). Our results show that after heavy PWN loading or hypoxic treatment, the thickness of ECM in the tracheal tubes of the vector beetle is remarkably enhanced. Regardless of an intricate relationship between ECM thickness and tubular diameter, the thickened ECM layer results in more elastic tubes that qualify a robust compression. Such improvement in compression capacity via tubular elasticity might be response to acute hypoxia or PWN loading, different from the chronic adaption that involve tracheal diametric expansion in *Drosophila* larvae incubated under constant hypoxic condition for generations (*Henry and Harrison, 2004*). In the PWN-beetle system, heavy PWN loading occupies luminal volume needed by tracheal ventilation for the beetle's regular respiration or intensive respiration during dispersal flight in the search of suitable host pine trees. The enhanced tracheal elasticity adapts vector insects to hypoxic stress allowing more oxygen inhalation and provides adequate lumen volume to carry large numbers of nematodes. Therefore, altered mechanical properties of the beetles' trachea after PWN loading likely maintain a balance between a limited fitness cost to beetles and high vector competence for effective transmission of PWN.

Along with enhanced tracheal elasticity in the vector beetle, our study also found remarkable global expression changes of aECM genes after PWN loading regardless of the termination of tracheal differentiation and maturation within the first 5 days after eclosion. For example, metalloproteinases degrading old cuticle and promoting apical membrane expansion (*Glasheen et al., 2010*), cadherins and integrins participating in signal transduction between aECM and tracheal cells (*Hayashi and*

*Kondo, 2018*; *Öztürk-Çolak et al., 2016*) are significantly upregulated after heavy PWN loading. This result suggests that PWN loading causes dramatic epithelium reorganization in tracheal tubes of vector beetles. Therefore, our study provides an example that the tracheal system is able to immediately respond to the environment by epithelium reorganization through changed aECM components. Although aECM is crucial to determine mechanical properties of tubular organs including trachea, the role of its molecular components remains largely unknown. Using gene manipulation, we further identified *Muc91C*, which encodes a resilin-like protein in aECM, as the key component responsible for the thickening of non-chitin aECM layer and the improved tracheal elasticity. However, due to incomplete reduced elasticity under hypoxic condition in ds*Muc91C*-injected beetles, other tracheal cellular components such as proteins embedded in the membrane may also regulate tracheal elasticity. Structurally, Muc91C in *M. alternatus* contains two types of repeat motifs but lacks chitin-binding RR2, a consensus sequence in resilin monomers (pro-resilins) (*Michels et al., 2016*), suggesting that its elasticity trait resides in the repeat motifs. Consistent with the absence of pro-resilin in the tracheal tubes of vector beetles, *Drosophila* larval foregut is devoid of pro-resilin, but contains resilin-like Cpr56F (*Lerch et al., 2020*). Collectively, these examples suggest that resilin-like proteins are employed instead of resilin monomers in deformation of various cuticle-free organs. In addition to the repeated motifs, water retention caused by the glycosylation of mucin (*Wagner et al., 2018*) may also promote elasticity, given that water coating may plasticize Mucin91C and provide additional elasticity from the surface tension of the liquid, as proved in spider silk (*Vollrath and Edmonds, 1989*). Interestingly, remarkably upregulated *Muc91C* expression was also detected in flight muscles and midgut of beetles with heavy PWN loading. Because highly glycosylated macromolecular mucins are known to protect cells against microbial infection (*Martens et al., 2018*), higher systematic *Muc91C* level in various tissues may contribute to maintaining the integrity of mucosal barriers to alleviate the detrimental effects of nematodes invasion, such as increased ROS reported in previous studies (*Zhou et al., 2018*).

This study found that hypoxia promotes tracheal elasticity by upregulating the expression of *Muc91C*, suggesting that *Muc91C* is related to hypoxia adaptation. Similarly, significant upregulated expression of pro-resilin has been demonstrated in an intertidal crustacean, the copepod *Tigriopus californicus*, when exposed to extremely low oxygen levels (*Graham and Barreto, 2019*). Therefore, the sensitivity of gene expression of aECM components in epithelial tissues to hypoxia is likely a pervasive phenomenon in arthropods. Clustering into a separate lineage in the mucin phylogenetic tree (*Figure 6—figure supplement 1*), all insect Muc91C proteins investigated in this study had no human homologues and were only distantly related to other mucins, suggesting their independent evolutionary origin in insects. This result is consistent with a previous study reporting that Muc91C is the only mucin structurally similar to pro-resilins among 23 mucins and mucin-related proteins in *D. melanogaster* (*Syed et al., 2008*). In Asian longhorned beetles (*Anoplophora glabripennis*), for instance, AgMuc91C has a 84% sequence similarity with *MaMuc91C* (*Figure 6*). Therefore, further examination of *Muc91C* expression patterns and its responses to hypoxia in other pine sawyer beetles closely related to our study species, such as *Monochamus saltuarius,* can help to assess the role of *Muc91C* in facilitating heavy PWN loading and thus to pine wood disease. Overall, our study provides evidence of exploitation of hypoxia-enhanced tracheal elasticity by a plant parasitic nematode to improve vector competence in this unique pathogen–vector system and offers new insights for the development of novel targets to control the pine wilt disease.

## Materials and methods
### Collection and rearing of nematodes and beetles
Mature larvae of sawyer beetles (*M. alternatus*) were collected from dead pine trees in the Fuyang area, Hangzhou City, Zhejiang Province, and reared in an incubator (26°C, humidity 35%) until they pupated or for 7 days after eclosion. *B.xylophilus* were originally acquired from infected trees in Shannxi Province and maintained with *Botrytis cinerea* in potato dextrose agar (BD, 213400) plates at 25°C for generations until used in the experiments.

### Nematode loading, tracheal dissections, and nematode quantification
50 ml Erlenmeyer flasks containing *B. cinerea* growing in autoclaved (121°C, 30 min) barley medium (10 g barley in 15 ml water) for 2 weeks at 25°C were inoculated with 1000 propagative nematodes.

After the nematodes had fed for 5 days, autoclaved pine wood sawdust layer was spread on top of the barley layer of the rearing flasks and cultured in 4°C for another 2 weeks. When the color of the pupae's eyes turned dark, they were put on the nematode-infected sawdust layer until 6 days after eclosion. Emerged adult beetles were dissected by ventral filleting, and the tracheal tubes located on both sides of the thorax were dissected and carefully deprived of surrounded muscle under a stereomicroscope (Olympus SZX16, Japan). The excised tracheal tubes and the left part of the body were washed in PBST (Solarbio, P1033) to irrigate the nematodes and presence of nematode was confirmed under a stereomicroscope (Olympus SZX16, Japan). The total number of isolated nematodes per beetle was counted in plates with an inverted microscope (Olympus CKX41, Japan).

## Measurement of oxygen concentrations

The measurement of oxygen concentrations followed previous studies with modifications (*Brune et al., 1995*; *Pettersen et al., 2005*). The Oxygen Microsensors with tip diameters of 25 µm (Unisense OX-25-905248, Denmark) and 100 µm (Unisense OX-100-909667, Denmark), respectively, were used to penetrate into trachea tubes (which are visible after ventral filleting and gut removal) and poke into the atrium cavity of the largest spiracles located on both sides of the thorax. This protocol was used for beetles without or with nematodes to obtain high-resolution profiles of oxygen concentrations. Prior to use, the electrodes of the microsensors were polarized overnight in deionized and continuously aerated water. Calibration was carried out before and after each experiment and was done by measuring the current when the microelectrode was placed in water saturated with air (21 kPa $O_2$) and measuring the background current in 1% $Na_2SO_3$ (0 kPa $O_2$). The values were measured with an amperemeter (Unisense oximeter-5591, Denmark) connected to a computer with a software SensorTrace Logger (Unisense, Denmark). Oxygen microsensor was fixed on a micromanipulator. Once placed into tracheal tubes and poking into the atrium cavity under a stereomicroscope, the microsensor was paused for at least 20 s to ensure sufficient data collection. The microsensor was then pulled out and recording was continued for 15 s to record values of the ambient air. To take into account temperature-derived fluctuation in saturated oxygen concentration of different tests, we converted the concentrations to its corresponding partial pressure, and the average partial pressure of oxygen during the 20 s inside the trachea was calculated as the final data.

## Hypoxic treatment

Hypoxic treatment was performed in a hypoxic chamber (FLYDWC-50; Fenglei Co., Ltd, China) placed in an artificial box to automatically control the ambient temperature, air flow, and pressure of oxygen ($PO_2$). Adult beetles without nematodes were placed in the ventilated chamber, in which air flow was balanced with pure nitrogen to achieve the required 1% $O_2$. The adult beetles were maintained in the chamber for 6 hr, 12 hr, or 24 hr at 25 ± 1°C in the dark.

## Measurement of rubber-like degree

Adult beetles post 6 days after eclosion were dissected by ventral filleting, and the morphological trait of thoracic tracheal tubes without and with heavy PWN were observed under a stereomicroscope (Olympus SZX16, Japan). Tracheal tube, which lost its original orange metallic luster and became pink and leathery, is defined as a rubber-like tube. The number of rubber-like tubes and total number of thoracic tracheal tubes oxygenate to flight muscle were counted and the rubberization degrees were set into three degrees according to the ratio of the former two values (Ⅰ, none; Ⅱ, less than 50% rubber-like tubes; Ⅲ, 50–100% rubber-like tubes). Then, the number of PWN harbored in beetles was counted.

## Measurement of mechanical properties with tensile testing

The two longest tracheal tubes linking the largest spiracle on both sides of the thorax and the spiracle basal to the head of the beetle were dissected as mentioned above and were maintained at room temperature (RT) in PBS (Solarbio, P1020). According to a previous study on muscle fibers (*Hakim et al., 2018*; *Kaiser et al., 2019*), tensile testing was performed with a micromechanical system (Aurora Scientific, ON, Canada) equipped with a length controller (322C), a 10 mN force transducer (405A), and dynamic control software suite DMC (ASI 600A). The measurements were performed inside a bath with PBS at room temperature. The dissected tracheal tubes were tied to the

fixed hooks at the ends of length controller and force transducer using nylon fibers. Prior to testing, the unstretched lengths ($L_0$) and radius ($r$) of the tubes were measured with caliper under an inverted microscope (Olympus CKX41, Japan). All samples were tested at a constant strain rate of 10% $L_0$/3 s until failure or 2×$L_0$ along the axial direction of tubes, as described in tensile testing on muscle fibers (*Krysiak et al., 2018*). The stiffness of the tubes was characterized by the Young's modulus ($E$), which is a constant specifying the elastic properties of a material (*Feynman et al., 1965*). The Young's modulus was determined from the linear region of the stress–strain curve, usually between 30% and 80% strain, using the equation of $E = \{(F/\pi r^2)/(\Delta L/L_0)\}$.

## Transmission electron microscopy of the tracheal tubes

The two longest tracheal tubes linking the largest spiracle on both sides of the thorax and the spiracle basal to head of the beetle were dissected and instantly fixed with 2.5% (vol/vol) glutaraldehyde (Coolaber, SL1790) and 4% (vol/vol) paraformaldehyde (Solarbio, P1112) with phosphate buffer (PB) (0.1 M, pH 7.4) (Macklin, H885798). The samples were then washed four times in PB and then tubes were postfixed with 1% (wt/vol) osmium tetraoxide in PB for 2 hr at 4°C, dehydrated through a graded ethanol series (30, 50, 70, 80, 90, 100%, 100%, 7 min each) into pure acetone (2×10 min). Samples were infiltrated in graded mixtures (3:1, 1:1, 1:3) of acetone and SPI-PON812 resin (16.2 g SPI-PON812, 10 g DDSA, and 8.9 g NMA), and then changed to pure resin. Finally, tubes were embedded in pure resin with 1.5% BDMA and polymerized for 12 hr at 45°C, 48 hr at 60°C. The ultrathin sections (70 nm thick) were sectioned with microtome (Leica EM UC6, Germany), double-stained by uranyl acetate and lead citrate, and examined by a transmission electron microscope (FEI Tecnai Spirit120kV, OR).

## RNA-seq and transcriptome analysis

Each sample for RNA-seq was prepared by combining four individual beetles (two females and two males, 6 days after emergence) for one library construction. The two libraries included trachea of sawyer beetles with and without PWN. Tracheal tubes were extirpated, washed with PBS to remove the nematodes, and immediately flash frozen with TRIzol (Invitrogen, 15596018) and stored in –80°C before use. Total RNA was isolated from the frozen samples using TRIzol reagent according to the manufacturer's protocol. mRNA was purified with a Dynabeads mRNA purification kit (Invitrogen, 61012). PolyA containing mRNAs was enriched with oligo (dT) magnetic beads (NEB, S1419S), fragmented with RNA fragmentation reagent, and subjected to the following procedure: first- and second-strand cDNA synthesis, purification, end repair, single-nucleotide addition, ligation of adapters, purification of ligated products, and PCR amplification for cDNA template enrichment. The RNA-seq library preparation kit for whole transcriptome discovery (compatible with Illumina, San Diego, CA) was used (Illumina Whole-Genome Gene Expression BeadChips). The 320–420 bp products were purified with the MiniElute gel extraction kit (QIAGEN, Hilden, Germany) and sequenced with the Illumina novaseq 6000 platform at Biomarker Technologies Co, LTD.

## Identification and analysis of aECM-related genes from the transcriptome and phylogenetic analysis

Using the SignalP 3.0 (http://www.cbs.dtu.dk/services/SignalP), we predicted secreted proteins in DEGs based on the presence and extension of signal peptides. The retrieved proteins were manually confirmed by Nr, GO, KO annotations, and using pfam and a BlastX search against the amino acid sequences of known ECM-related genes from flybase (http://flybase.org/). Transmembrane domains were analyzed by TMHMM server v. 2.0 (http://www.cbs.dtu.dk/services/TMHMM/). The RR2 Consensus sequences were predicted using the cuticleDB website (http://bioinformatics2.biol.uoa.gr/cuticleDB/index.jsp). The O-glycosylation sites of mucin proteins were determined by NetOGlyc 4.0 (https://www.cbs.dtu.dk/services/NetOGlyc). To verify the classification of resilin-related proteins and other mucin proteins, phylogenetic analysis was performed using the full-length amino acid sequences of representative proteins from other species retrieved from the GenBank database. Multiple alignments of the amino acid sequences were performed using the ClustalW program in MEGA 10.0. The phylogenetic trees were constructed adopting neighbor-joining (NJ) method using 1000 bootstrap repetitions. All resilin-related and mucin sequences of other model species used in this study are provided as *Figure 4—source data 1* and *Supplementary file 1*.

## Real-time PCR

Total RNA was extracted from thoracic tracheal tubes of each beetle in different treatments using TRIzol reagent according to the manufacturer's protocol and stored at –80°C until used in experiments. The total number of nematodes in corresponding tracheal samples were counted. RNA quality was assessed with an ND-1000 spectrophotometer (NanoDrop Technologies, DE). About 1 µg of RNA was used as template to produce cDNA with FastQuant RT kit with a genomic DNA eraser (Tiangen, KR106). RT-qPCR was performed using Talent qPCR PreMix (SYBR Green) (Tiangen, FP209) and on a 7300Plus Real-Time PCR system (Applied Biosystems, MA). At least five biological replicates were assayed for statistical analysis. The amount of mRNA was calculated by the change in cycle threshold ($\Delta\Delta Ct$) method and was normalized to control rp49 mRNA values. qPCR primers are listed in **Supplementary file 2**.

## Western blot analysis

Total proteins were extracted from thoracic tracheal tubes of each beetle in different treatments using TRIzol reagent according to the manufacturer's protocol and stored at –80°C until used in experiments. The protein extracts (100 mg) were electrophoresed on 4–12% SDS-PAGE precast gels (EASYBIO, China) and then transferred to 0.22 µm polyvinylidene difluoride (PVDF) membranes (Millipore). The membrane was incubated with polyclonal antibody against target protein (anti-Muc91C, developed by BGI, China, 1:4000). Goat anti-rabbit IgG (EASYBIO, 1:5000) was used as the secondary antibody. Monoclonal antibody against Histone H3 (EASYBIO, 1:2000) was used as an internal control. Goat anti-mouse IgG (EASYBIO, 1:5000) was used as the secondary antibody. Protein bands were detected by SuperSignal West Atto Ultimate Sensitivity Substrate (Invitrogen, A38554).

## Whole-mount immunohistochemistry

Thoracic tracheal tubes of each beetle in different treatments were dissected and instantly fixed with 4% paraformaldehyde overnight. After being washed with 1× PBS buffer, the samples were permeabilized with 0.5% TritonX-100/PBS at 37°C for 20 min, then blocked with 5% goat serum for 1 hr, and incubated with affinity-purified polyclonal rabbit antibody against Muc91C (produced by BGI, China, 1: 100) at 4°C for 24 hr. The samples were then washed three times for 5 min each with 0.5% TritonX-100/PBS, Alexa Fluor-488 goat anti-rabbit IgG (Invitrogen, A11008, 1: 500) was used as the secondary antibody. The cellular nucleus was stained with Hoechst 33342 (Invitrogen H3570, 1:1000) for 20 min. The cell membrane system was stained with CellTracker CM-DiI dye (Invitrogen, C7001, 1: 500) for 30 min. Finally, the samples were mounted with Fluoroshield mounting medium (Solarbio, S2100). Fluorescence was detected using an confocal laser-scanning microscope (SP8 Lightning, Leica). Taenidial folds are autofluorescence excited by 488 nm or 561 nm lasers and have higher autofluorescence excitated by 561 nm laser. Images were processed using LAS X Navigator (Leica).

## RNA interference

Double-stranded RNAs (dsRNAs) were synthesized with cDNA templates possessing T7 RNA polymerase promoter sequences on both ends (**Supplementary file 2**), according to the T7 RiboMax Express RNAi System Kit (Promega, P1700) protocol. Within 2 days after molting, approximately 60 adults beetle were injected at the joint of the second and the third abdominal segment with 30 µg of dsRNA (<5 µl) using a 7000-series modified microliter syringe (Hamilton, Bonaduz, Switzerland). Control beetles were injected with dsGFP as a negative control. The injected adult beetles were kept in standard conditions for 4 days, then treated with 1% $O_2$ for 12 hr. The injected adult beetles used in the nematodes loading trial were kept in 50 ml Erlenmeyer flasks with barley medium containing dispersal $L_{III}$ and $L_{IV}$ nematodes at 25°C for 5 days. When the adults were 7 days old, they were dissected to obtain trachea samples for the different assays, and disposed as described earlier.

## Statistical analysis

The sample size is determined according to previous publications in the pine sawyer beetle species (**Wu et al., 2019**; **Zhao et al., 2016**; **Zhou et al., 2018**). For nematode loading, hypoxic treatments and RNAi treatments, individuals were randomly allocated into experimental group and control group, and no restricted randomization was applied. For nematode quantification, measurements of mechanical properties with tensile testing, researchers were blind to RNAi or hypoxic treatments

before counting or measurements. All experiments were performed for at least three independent biological replicates.

A Pearson correlation coefficient was used to measure the correlation between PWN loading and partial pressure of oxygen. The semi-log fit using least square method was chosen for curve fitting. The fit formula was $Y = 10.55 + (-9.629) \times \log(X)$ with an $R^2$ value of 0.85. A Kolmogorov–Smirnov test was used to check the normality of data. When the assumption of normal distribution was met, Student's $t$-test and one-way ANOVA with Dunn's or Tukey's multiple comparison were used for two-group and three-group comparisons, respectively. When the assumption of normal distribution was not met, Mann–Whitney U test and Kruskal–Wallis with Dunn's multiple comparison were used for two-group and three-group comparisons, respectively. Statistical analysis was performed with GraphPad Prism 8.0 (GraphPad software, San Diego, CA).

## Acknowledgements

This work was funded by the National Natural Science Foundation of China (32088102, 32061123002, 32230066 and 31970466) and National Key Research and Development Program of China (2021YFC2600100). We are grateful to Yujia Xiang and Jin Ge for the assistance in bioinformatic analysis. We thank Kathryn E Bushley for reviewing the manuscript.

## Additional information

### Funding

| Funder | Grant reference number | Author |
| --- | --- | --- |
| National Natural Science Foundation of China | 32088102 | Jianghua Sun |
| National Natural Science Foundation of China | 32061123002 | Jianghua Sun |
| National Key Research and Development Program of China | 2021YFC2600100 | Jianghua Sun |
| National Natural Science Foundation of China | 32230066 | Lilin Zhao |
| National Natural Science Foundation of China | 31970466 | Jiao Zhou |

The funders had no role in study design, data collection and interpretation, or the decision to submit the work for publication.

### Author contributions

Xuan Tang, Conceptualization, Data curation, Formal analysis, Investigation, Visualization, Methodology, Writing – original draft, Writing – review and editing; Jiao Zhou, Conceptualization, Data curation, Funding acquisition, Investigation, Methodology, Writing – review and editing; Tuuli-Marjaana Koski, Validation, Writing – review and editing; Shiyao Liu, Investigation, Visualization; Lilin Zhao, Conceptualization, Funding acquisition, Writing – review and editing; Jianghua Sun, Conceptualization, Supervision, Funding acquisition, Investigation, Writing – review and editing

### Author ORCIDs

Jianghua Sun http://orcid.org/0000-0002-9465-3672

### Decision letter and Author response

Decision letter https://doi.org/10.7554/eLife.84621.sa1
Author response https://doi.org/10.7554/eLife.84621.sa2

## Additional files

### Supplementary files
- Supplementary file 1. Species and GenBank accession no. of pro-resilins and mucins.
- Supplementary file 2. Primers for real-time qPCR and RNAi experiments.
- MDAR checklist

### Data availability

All data generated or analyzed during this study are included in the manuscript and supporting file. Source data files have been provided for Figures 1, 2, 3, 4, 6 and 7. The RNA-seq data reported in this study was deposited in the Genome Sequence Archive (Genomics, Proteomics & Bioinformatics, 2017) in National Genomics Data Center (Nucleic Acids Research, 2020), Beijing Institute of Genomics (China National Center for Bioinformation), Chinese Academy of Sciences, under accession number CRA006464 and CRA008617 that are publicly accessible at https://ngdc.cncb.ac.cn/gsa/browse/CRA006464 and https://ngdc.cncb.ac.cn/gsa/browse/CRA008617.

The following dataset was generated:

| Author(s) | Year | Dataset title | Dataset URL | Database and Identifier |
|---|---|---|---|---|
| Tang X | 2023 | RNA-seq of pine sawyer beetle with PWN | https://ngdc.cncb.ac.cn/gsa/browse/CRA006464 | Genome Sequence Archive, CRA006464 |

The following previously published dataset was used:

| Author(s) | Year | Dataset title | Dataset URL | Database and Identifier |
|---|---|---|---|---|
| Tang X | 2022 | RNA-Seq of tracheal tubes in adult pine sawyer beetle | https://ngdc.cncb.ac.cn/gsa/browse/CRA008617 | Genome Sequence Archive, CRA008617 |

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
