## [Editor Report]

This valuable work explores how pathogens can cause hypoxia in insect vectors and how responses to hypoxia can be exploited to promote vector competence. Using Pine Wood Nematode (PWN) infection of pine sawyer beetles the authors demonstrate that PWN loading activates hypoxia in the vector's tracheal system. The data were collected and analyzed using solid and validated methodology.

---

## [Decision Letter]

**Decision letter after peer review:**

[Editors’ note: the authors submitted for reconsideration following the decision after peer review. What follows is the decision letter after the first round of review.]

Thank you for submitting the paper "Hypoxia-induced tracheal elasticity in vector beetle facilitates the loading of pinewood nematode" for consideration by *eLife*. Your article has been reviewed by 3 peer reviewers, including Sofia J Araujo as the Reviewing Editor and Reviewer #1, and the evaluation has been overseen by a Senior Editor.

Comments to the Authors:

We are sorry to say that, after consultation with the reviewers, we have decided that this work will not be considered further for publication by *eLife*.

Specifically, the reviewers find that although the work is novel and of relevance it lacks strong experimental backup. The work would gain from deeper genetic and cell biological analysis and we hope you will find the reviewers' comments useful.

*Reviewer #1 (Recommendations for the authors):*

Tang et al. investigated if pathogen loading can increase hypoxia in the insect vector's tracheal system. To do so, the authors use pine wood nematodes (PWN) and their infection of Monochamus beetles via their tracheal system. They find that the number of the nematodes in the trachea is negatively correlated with the oxygen level in the tracheal system of its vector beetle, leading to hypoxia. They also observe that PWN loading and hypoxia enhanced the elasticity of tracheal tubes and increased the thickness of tracheal aECM. Analysing the transcriptome they identify changes in aECM related genes in infected beetles. They find that, upon infection, Muc91C has the highest increase in expression and that its downregulation in beetles reduces PWN loading and tracheal elasticity.

Strengths:

The analysis of the pathogen's strategy to maintain itself in its vector by modulating the vector's hypoxic responses. The observation that nematode entry decreases oxygen levels, inducing hypoxia, which in turn increases tracheal tube elasticity by Muc91C upregulation.

The demonstration that heavy PWN loading, or hypoxic treatment, enhances the elasticity of the tracheal tubes of the vector beetle.

The molecular analysis of the tracheal response to infection and aECM gene expression changes.

Weaknesses:

The molecular analysis of Muc91C in the tracheal system of the beetles would gain with a more detailed cellular observations.

This work clearly demonstrates the effect of PWN loading on the elasticity of tracheal tubes. However, its major weakness lies on the lack of a detailed analysis of Muc91C expression and localization.

Enhanced tracheal elasticity induced by hypoxia upon infection relies on the expression of Muc91C. Where does Muc91C localize in relation to the chitinous aECM? Are the effects observed due to protein levels?

I understand that these experiments may be difficult to perform in Monochamus beetles, but the authors could try to find answers to these questions using *Drosophila melanogaster*.

*Reviewer #2 (Recommendations for the authors):*

The manuscript by Xuan Tang and colleagues contains the experimental analyses of the relationship between the pine sawyer beetle and the pinewood pathogen nematode (pwn). The authors find that the load of pwn in the beetle correlates with a hypoxia response, in turn inducing the expression of the mucin Muc91C, a presumptive cuticular protein, in the tracheal system. They claim that the underlying mechanism involves hypoxia-induced Muc91C-dependent changes in tracheal cuticle elasticity. Other mucins do not seem to be implicated in this process. The circuitry nematode-load, hypoxia, Muc91C-expression and increased elasticity ends with the release of the nematodes on pines, where they act as a pest.

The addressed biological, ecological problem is really exciting. The model of the sequence of events is intriguing and, partly, backed up with solid data, also from previous works (Wu et al., 2019 cites in the manuscript).

Besides some minor points that I will mention below, there is a major one that puzzles me. It is about the relationship between tracheal development/differentiation, gene expression (among others of Muc91C) and the analyses of beetles in different experiments (hypoxia, transcriptome etc). For instance, nematodes loading on pupae is described on lines 450ff: when the "colour of.. eyes turned dark, nematodes were added": which tracheal stage is this? Changes in Muc91C expression in hypoxia assays and transcriptomics were then performed on adult beetles, when tracheal development/differentiation is supposed to be largely terminated, at least in other insects. Hence, how can Muc91C alter tracheal physics after the overall architecture of the tracheae is established? One possibility is that I am wrong and the tracheal development/differentiation continues also in adult beetles even seven days post eclosion (as stated in the Materials and methods section). Another possibility is that newly forming tracheal branches/tips respond to hypoxia and nematode loads, but not portions of the tracheae established already during metamorphosis.

A related problem is the equation of elasticity of the tracheal cuticle and oxygen concentration: is there any logical argument that allows us to suppose that a more elastic extracellular matrix might facilitate oxygen diffusion? According to the authors, the tracheal cuticle becomes even thicker as a response to nematode loading and hypoxia. If I remember well, hypoxia-incubated *Drosophila* larvae tend to produce tracheal tubes with a larger diameter (after moulting, not as an immediate response) for better aeration. A subtle problem arises also, when considering Muc91C incorporation into the cuticle and cuticle property changes after tracheal development/differentiation: the tracheal cuticle formed at metamorphosis would have a stiffer "more normal" consistency than the "newly" added cuticle: the old cuticle would restrict the elasticity of the new one like a concrete wall the elasticity of a balloon. Again, as mentioned above: we need a more detailed description of the course of tracheal development/differentiation in relation to gene expression (Muc91C).

Another point concerns the description of putative elastic proteins in insects: Muc91C contains repeats that may be responsible for elasticity. However, this has not been shown (while it has been shown for Resilin). The authors should be careful in this regard. Muc91C as a mucin is decorated by sugars that may bind water, this swelling may confer "elasticity" to the cuticle rather than its repeats. Increased water content in the cuticle may also be responsible for "thicker" cuticles in nematode loaded or hypoxic tracheae: increased thickness may be an artefact of preparation. NB: the two TEM images in Figure 5 C do not have comparable quality. Preparation, the thickness of the sections or contrasting differ too much.

A problem might also be the function of Muc91C: first, can the authors exclude that it is not expressed in tissues associated with the tracheal system when transcriptomics were carried out? In *Drosophila*, the Muc91C expression is detected in the nervous system and the fat body. Second: RNAi against Muc91C seems to produce viable beetles. I would expect a reduction in overall fitness, though. May this, if the case, affect the interaction with pwn?

The cloudy situation of the relationship tracheal development/differentiation, gene expression and organ response should be clarified: we need a thorough description of tracheal development/differentiation; the course of tracheal development/differentiation should be next correlated with gene expression (for instance Muc91C). This is a crucial experiment as this would be the first example that the main tracheal system may respond to the environment (here pwn and hypoxia) even after differentiation. Moreover, it would strengthen or serve to refute the argument of Muc91C being expressed and functional beyond terminal differentiation of the tracheal system. Overall, I consider this issue as very crucial as the current version reports only on a phenomenon rather than on a biological process.

The robustness of their conclusion on the relevance of Muc91C function would increase if additional genes (only mucs are not sufficient) expression were knocked-down (such as the chitin synthase coding gene on the list in Figure 3) and the resulting phenotypes analysed.

We need also clarification regarding the rubberization issue (lines 163ff): it is unclear how this was measured. The description in the material and methods section and in the figure legend are insufficient.

*Reviewer #3 (Recommendations for the authors):*

Tang and colleagues demonstrated the importance of hypoxia-mediated enhancement of tracheal elasticity in vector beetle M. alternatus to transport more nematodes to new host pine trees. Tracheal infection by PWN provoked hypoxia inside the trachea and this made the host tracheal aECM thicker than the non-infected host, increasing tracheal elasticity. The authors then revealed the significance of Muc91C gene, which encodes a resilin-like mucin protein and is highly up-regulated upon nematode infection, by RNAi experiment. Indeed, tracheal aECM was not developed (not thickened) in the Muc91C-RNAi insect and it's elasticity was significantly lower than control insect. Interestingly, when the development of tracheal aECM was disrupted by Muc91C-RNAi, the total amount of trachea-residing PWN was significantly decreased, indicating that PWN-mediated tracheal hypoxia is critical to vector competence to carry more nematodes.

Although the authors have come up with fairly clear results, there are some shortcomings. A key point of this paper is the elasticity and thickness of aECM of tracheal tissue, but they just showed TEM images and mathematical constant (Young's modulus) to demonstrate them. This is ambiguous and very insufficient to interpret the importance of tracheal elasticity. The author should present more rigid and scientific results to explain this key context (e.g., at least need to do immunostaining of resilin-like mucin protein between PWN-infected and non-infected insects or other measuring methods).

Nevertheless, they found a novel functional gene (Muc91C), which is partially important to tracheal development such as thickness and elasticity.

[Editors’ note: further revisions were suggested prior to acceptance, as described below.]

Thank you for resubmitting your work entitled "Hypoxia-induced tracheal elasticity in vector beetle facilitates the loading of pinewood nematode" for further consideration by *eLife*. Your revised article has been evaluated by Dominique Soldati-Favre (Senior Editor) and a Reviewing Editor.

The manuscript has been improved but there are some remaining issues that need to be addressed, as outlined below:

1) The authors show that Muc91C localizes underneath the chitinous aECM in tracheal tubes upon infection. This is a very good result, however, the images are not clear and the manuscript would gain from higher magnification images focusing on a small section of the membrane and showing the tracheal cells in more detail.

2) Reference to the elastic property of repeat A: the way the authors state that Muc91 repeat A has been shown to "regulate elasticity" is, to my understanding, wrong. First, repeat A from pro-resiling has been shown to confer elasticity, not of Muc91. Second, elasticity does not come from the repeats, but from their ability to form di-tyrosine bridges, and this has not been shown.

3) Young's modulus: did I understand correctly that the forces were measured along the main axis of the tracheal tubes? To me, they should be measured perpendicular to the main axis. Please describe the method more in detail. Forces may be different in different directions.

*Reviewer #1 (Recommendations for the authors):*

The authors show that Muc91C localizes underneath the chitinous aECM in tracheal tubes upon infection. This is a very good result, however, the images are not clear and the manuscript would gain from higher magnification images focusing on a small section of the membrane and showing the tracheal cells in more detail.

*Reviewer #2 (Recommendations for the authors):*

– Reference to the elastic property of repeat A: the way the authors state that Muc91 repeat A has been shown to "regulate elasticity" is, to my understanding, wrong. First, repeat A from pro-resiling has been shown to confer elasticity, not of Muc91. Second, elasticity does not come from the repeats, but from their ability to form di-tyrosine bridges, and this has not been shown.

– My argument that Muc91 may be involved in retaining water in the cuticle and thereby contributing to elasticity was refuted. Fine with me, if this was done with a valid argument. Muc91 as a mucin may bind water and thereby contribute to elasticity, and not (only) via its repeats.

– Young's modulus: did I understand correctly that the forces were measured along the main axis of the tracheal tubes? To me, they should be measured perpendicular to the main axis. Please describe the method more in detail. Forces may be different in different directions.

---

## [Author Response]

[Editors’ note: the authors resubmitted a revised version of the paper for consideration. What follows is the authors’ response to the first round of review.]

Reviewer #1 (Recommendations for the authors):Tang et al. investigated if pathogen loading can increase hypoxia in the insect vector's tracheal system. To do so, the authors use pine wood nematodes (PWN) and their infection of Monochamus beetles via their tracheal system. They find that the number of the nematodes in the trachea is negatively correlated with the oxygen level in the tracheal system of its vector beetle, leading to hypoxia. They also observe that PWN loading and hypoxia enhanced the elasticity of tracheal tubes and increased the thickness of tracheal aECM. Analysing the transcriptome they identify changes in aECM related genes in infected beetles. They find that, upon infection, Muc91C has the highest increase in expression and that its downregulation in beetles reduces PWN loading and tracheal elasticity.Strengths:The analysis of the pathogen's strategy to maintain itself in its vector by modulating the vector's hypoxic responses. The observation that nematode entry decreases oxygen levels, inducing hypoxia, which in turn increases tracheal tube elasticity by Muc91C upregulation.The demonstration that heavy PWN loading, or hypoxic treatment, enhances the elasticity of the tracheal tubes of the vector beetle.The molecular analysis of the tracheal response to infection and aECM gene expression changes.

Thanks for your positive comments.

Weaknesses:The molecular analysis of Muc91C in the tracheal system of the beetles would gain with a more detailed cellular observations.

This comment is really helpful. In this revised version, we conducted immunostaining of Muc91C in the tracheal system of the beetles. We found heavy nematode loading resulting in substantial upregulation of Muc91C beneath the layer of taenidial folds and above the apical membrane of the tracheal cell, as shown in line 290-295 and Figure 4E & F. This result is consistent with the nematode-thickened aECM layer showed in TEM, thus strongly demonstrating that Muc91C is the component of the PWN-thickened aECM layer showed in TEM.

This work clearly demonstrates the effect of PWN loading on the elasticity of tracheal tubes. However, its major weakness lies on the lack of a detailed analysis of Muc91C expression and localization.Enhanced tracheal elasticity induced by hypoxia upon infection relies on the expression of Muc91C. Where does Muc91C localize in relation to the chitinous aECM? Are the effects observed due to protein levels?I understand that these experiments may be difficult to perform in Monochamus beetles, but the authors could try to find answers to these questions using *Drosophila melanogaster*.

This comment is essentially the same with Comment #1 above.

To examine Muc91C expression at the protein level, we carried out Western blot and found nematode loading significantly enhance Muc91C in the trachea of beetles, as shown in line 285-290 and Figure 4D. By performing immunostaining using Muc91C antibody, we found Muc91C were expressed beneath the layer of chitinous taenidial folds and above the apical membrane of the tracheal cell, as shown in line 290-295 and Figure 4E & 4F. This is consistent with the thickened aECM layer in TEM analysis. Therefore, we confirm that the observed effects are due to protein levels thus revealing a strong association between elasticity and the expression of Muc91C.

Reviewer #2 (Recommendations for the authors):The manuscript by Xuan Tang and colleagues contains the experimental analyses of the relationship between the pine sawyer beetle and the pinewood pathogen nematode (pwn). The authors find that the load of pwn in the beetle correlates with a hypoxia response, in turn inducing the expression of the mucin Muc91C, a presumptive cuticular protein, in the tracheal system. They claim that the underlying mechanism involves hypoxia-induced Muc91C-dependent changes in tracheal cuticle elasticity. Other mucins do not seem to be implicated in this process. The circuitry nematode-load, hypoxia, Muc91C-expression and increased elasticity ends with the release of the nematodes on pines, where they act as a pest.The addressed biological, ecological problem is really exciting. The model of the sequence of events is intriguing and, partly, backed up with solid data, also from previous works (Wu et al., 2019 cites in the manuscript).

Thanks for your positive comments.

It is about the relationship between tracheal development/differentiation, gene expression (among others of Muc91C) and the analyses of beetles in different experiments (hypoxia, transcriptome etc). For instance, nematodes loading on pupae is described on lines 450ff: when the "colour of.. eyes turned dark, nematodes were added": which tracheal stage is this? Changes in Muc91C expression in hypoxia assays and transcriptomics were then performed on adult beetles, when tracheal development/differentiation is supposed to be largely terminated, at least in other insects. Hence, how can Muc91C alter tracheal physics after the overall architecture of the tracheae is established? One possibility is that I am wrong and the tracheal development/differentiation continues also in adult beetles even seven days post eclosion (as stated in the Materials and methods section). Another possibility is that newly forming tracheal branches/tips respond to hypoxia and nematode loads, but not portions of the tracheae established already during metamorphosis.

We totally agree with this comment that analysis of the relationship between gene expression and tracheal development would provide basic knowledge underlying this study. Actually, there is a fascinating developmental synchronization between PWN and the vector beetle in previous studies (Zhao et al., 2014) and the life cycles of the two species are shown in Figure 7. The formation of dispersal L_IV_ exclusively induced by late pupa beetle with black eyes and newly molted adult beetle (Zhao et al., 2013). Thus, the PWN loading to tracheal system of the beetle occurs after beetle eclosion. In parallel with this study, we have actually investigated the morphological and transcriptomic changes of tracheal development during metamorphosis of the vector beetle morphological observation has shown that, the pattern of tracheal network has been already formed at the late pupal stage (the time when propagative PWNs were introduced in the present study). Tracheal development continues within the first five days post eclsion and is almost terminated after five days post eclosion, given the dramatic changes of tracheal gene expression occurring within five days after eclosion. These results have been submitted to BioRxiv (Tang et al., 2022). Therefore, developmental pattern observed in vector beetles support the second possibility in this comment that newly forming tracheal branches/ tips respond but not portions of the trachea established already during *metamorphosis*. In the current study, we added analysis of the temporal changes for the gene expression of the 45 aECM-related genes (including *Muc91C*) at different time points post eclosion, using the released RNA-seq data as shown in line 251-261 and Figure 3—figure supplement 1. We found that all genes were expressed at low levels after five days post-elcosion, suggesting a termination of tracheal development before the transcriptome and hypoxia treatment were carried out.

A related problem is the equation of elasticity of the tracheal cuticle and oxygen concentration: is there any logical argument that allows us to suppose that a more elastic extracellular matrix might facilitate oxygen diffusion? According to the authors, the tracheal cuticle becomes even thicker as a response to nematode loading and hypoxia. If I remember well, hypoxia-incubated *Drosophila* larvae tend to produce tracheal tubes with a larger diameter (after moulting, not as an immediate response) for better aeration.

The facilitation of a more elastic ECM on oxygen diffusion is likely due to its role on tracheal ability to compress. In the revised text, we interpreted the association between aECM and oxgen diffusion in Introduction (line 75-96). In insects, including beetles, the tracheal system exhibits rapid cycles of compression and expansion, analogous to the inflation and deflation of vertebrate lungs. The tracheal volumes changed by this behavior increase the internal pressure for improvement of air convection and gases exchange, thus facilitating oxygen diffusion (Westneat et al., 2003). In addition, hypoxia increases the frequency of this behavior to promote gas exchange (Greenlee et al., 2013). There is evidence showing that the compression capacity is largely determined by tracheal mechanical properties. In the American cockroaches, the chitin structures in ECM layer are responsible for mechanical features that enable volume changes during respiration (Webster et al., 2011). Similarly, high inflation of lungs is related to the mechanical forces provided by ECM components (Berg et al., 1997; Wirtz and Dobbs, 2000).

In the revised text, we discussed the thickened tracheal cuticle in response to nematode loading and hypoxia by adding “Regardless of an intricate relationship between ECM thickness and tubular diameter, the thickened ECM layer results in more elastic tubes that qualify a robust compression. Such improvement of compression capacity via tubular elasticity might be response to acute hypoxia or PWN loading, different from the chronic adaption that involve tracheal diametric expansion in *Drosophila* larvae incubated under constant hypoxic condition for generations (Henry and Harrison, 2004).” in Discussion (line 449-455).

A subtle problem arises also, when considering Muc91C incorporation into the cuticle and cuticle property changes after tracheal development/differentiation: the tracheal cuticle formed at metamorphosis would have a stiffer "more normal" consistency than the "newly" added cuticle: the old cuticle would restrict the elasticity of the new one like a concrete wall the elasticity of a balloon. Again, as mentioned above: we need a more detailed description of the course of tracheal development/differentiation in relation to gene expression (Muc91C).

To address the concern about tracheal plasticity, we explained

PWN-induced epithelium reorganization in the Discussion (line 463-473). As mentioned above, tracheal development is terminated within the first five days after eclosion. However, PWN loading causes substantial expressional changes of 251 ECM related genes, suggesting dramatic epithelium reorganization in tracheal tubes of vector beetles. For example, in parallel with the increased expression of Muc91C, metalloproteinases degrading old cuticle and promoting apical membrane expansion (Glasheen et al., 2010), cadherins and integrins participating in signal transduction between aECM and tracheal cells (Hayashi and Kondo, 2018; Öztürk-Çolak et al., 2016) are significantly upregulated as well. Therefore, aECMs are already well-developed during metamorphosis and undergo reconstruction after PWN loading, resulting in a more elastic tracheal cuticle structure for supporting respiration.

Another point concerns the description of putative elastic proteins in insects: Muc91C contains repeats that may be responsible for elasticity. However, this has not been shown (while it has been shown for Resilin). The authors should be careful in this regard. Muc91C as a mucin is decorated by sugars that may bind water, this swelling may confer "elasticity" to the cuticle rather than its repeats. Increased water content in the cuticle may also be responsible for "thicker" cuticles in nematode loaded or hypoxic tracheae: increased thickness may be an artefact of preparation.

We have clarified the causal links between Muc91C and elasticity in line 350-356. Muc91C of the beetle contains two series of short repeated motifs located in the N-terminal (A repeats) and C-terminal (B repeats) region and the A repeats are shared by Muc91C homologues of other insects, as shown in Figure 5 of the current version of manuscript. The direct contribution of repeat A in elastic properties has been confirmed in several studies (Elvin et al., 2005; Lyons et al., 2007; Nairn et al., 2008), demonstrating that synthetic peptide chains containing either the A-repeat-containing region from *D. melanogaster* pro-resilin or a series of the consensus sequence, AQTPSSQYGAP, from the *A. gambiae* Muc91C (AGAP002367-PA) form rubberlike elastic materials. Actually, in a seminal paper about resilin-like gene products in insects (Andersen, 2010), DmMuc91C and AgaMuc91C are both classified as a resilin-like protein due to their abundance in short repeats containing proline and glycine, a common feature of proteins with long-range elasticity. In the revised version, we also carried out immunostaining of Muc91C as shown in line 290-295 and Figure 4E & 4F. We detected strong signals of Muc91C beneath the layer of taenidial folds and above the apical membrane of the tracheal cell, forming a continuous layer in PWN-loaded trachea. Thus, this result indicates that the enhanced expression of Muc91C instead of increased water content accounts for the thickened ECM observed by TEM.

NB: the two TEM images in Figure 5 C do not have comparable quality. Preparation, the thickness of the sections or contrasting differ too much.

The samples for were treated and prepared according to the same protocol. All the ultrathin sections share the same thickness (70nm), as shown in line 603. We have adjusted the contrasting of the TEM images in Figure 6A as suggested in this revised version.

A problem might also be the function of Muc91C: first, can the authors exclude that it is not expressed in tissues associated with the tracheal system when transcriptomics were carried out? In *Drosophila*, the Muc91C expression is detected in the nervous system and the fat body.

This concern makes sense. To exclude the influence of related tissues, we carefully removed the surrounded muscle under microscope when sampling the trachea for transcriptomes. These details of operation have been added in the Materials and methods in line 531. To elucidate the tissue specificity of *Muc91C*, we conducted qPCR for *Muc91C* expression in tracheal tubes, flight muscle and midgut. Compared to the other two tested tissues, *Muc91C* expression was mostly highly expressed in the trachea of both infested or non-infested adult beetles. These results have been added in line 280-285 and Figure 4C. In addition, the immunostaining of Muc91C has shown its primary localization in tracheal tubes but not in other associated tissues, as demonstrated by Figure 4E & 4F.

Second: RNAi against Muc91C seems to produce viable beetles. I would expect a reduction in overall fitness, though. May this, if the case, affect the interaction with pwn?

The viability of beetles was not affected by RNAi against *Muc91C* and we add this results in this revised vision. In RNAi efficiency assessment, TEM imaging and tensile testing, ds*GFP* or ds*Muc91C* were injected into beetle adults within two days after molting and treated with 1% O_2_ for 12h after dsRNA injection. All treated beetles survived until tracheal dissection, as shown in line 377 and Figure 6—figure supplement 1B. In the RNAi experiments for PWN loading, *dsGFP* and *dsMuc91C* resulted in similar proportions of death before dissection for counting, as shown in line 387-388 and Figure 6—figure supplement 1C.

Collectively, RNAi against *Muc91C* did not reduce the fitness of beetles.

The cloudy situation of the relationship tracheal development/differentiation, gene expression and organ response should be clarified: we need a thorough description of tracheal development/differentiation; the course of tracheal development/differentiation should be next correlated with gene expression (for instance Muc91C). This is a crucial experiment as this would be the first example that the main tracheal system may respond to the environment (here pwn and hypoxia) even after differentiation. Moreover, it would strengthen or serve to refute the argument of Muc91C being expressed and functional beyond terminal differentiation of the tracheal system. Overall, I consider this issue as very crucial as the current version reports only on a phenomenon rather than on a biological process.

This comment is very constructive and essentially the same with the comment #1. By thoroughly assessing the time course of metamorphosis, our work (Tang et al., 2022) has confirmed that the tracheal development/differentiation is almost terminated within the first five days after eclosion. As the comment suggested, we analyzed the temporal changes for the gene expression of the 45 ECM-related genes (including *Muc91C*) at different time points post eclosion, using the released RNA-seq data. In the revised text, we add this analysis in line 251-261 and Figure 3—figure supplement 1. We found that all genes were expressed at low levels after five days post-elcosion. For *Muc91C* specifically, its expression was 2-fold upregulated three days after eclosion but was downregulated during the later stages post eclosion. These results are consistent with the dynamics of other tracheal genes exhibiting upregulation mainly during the first five days post eclosion. Therefore, the current study provides an example that the tracheal system is able to respond to the environment. Moreover, PWN loading results in a 7-fold increase of *Muc91C* expression, a higher fold-change than that during tracheal development. This result thus supports the argument of Muc91C being expressed and functional beyond terminal differentiation of the tracheal system.

The robustness of their conclusion on the relevance of Muc91C function would increase if additional genes (only mucs are not sufficient) expression were knocked-down (such as the chitin synthase coding gene on the list in Figure 3) and the resulting phenotypes analysed.

Further studies should be carried out to systematically assess the function of diverse ECM components in nematode-altered tracheal elasticity as suggested. In this study, we focused on Muc91C, the foremost component that directly linked with the hypoxia- or PWN enhanced thickness and elasticity of the non-chitinous ECM layer on the basis of TEM observation, immunostaining and gene structure analysis. The incomplete reduced elasticity under hypoxic condition in ds*Muc91C*injected beetles suggested complementary roles of other components. In the revised text, we discussed the coordination among Muc91C and other ECM in determining tracheal elasticity in the Discussion (line 463-473).

We need also clarification regarding the rubberization issue (lines 163ff): it is unclear how this was measured. The description in the material and methods section and in the figure legend are insufficient.

Clarification regarding the rubberization issue has been added in the Materials and methods section in line 566-575, as suggested.

Reviewer #3 (Recommendations for the authors):Tang and colleagues demonstrated the importance of hypoxia-mediated enhancement of tracheal elasticity in vector beetle M. alternatus to transport more nematodes to new host pine trees. Tracheal infection by PWN provoked hypoxia inside the trachea and this made the host tracheal aECM thicker than the non-infected host, increasing tracheal elasticity. The authors then revealed the significance of Muc91C gene, which encodes a resilin-like mucin protein and is highly up-regulated upon nematode infection, by RNAi experiment. Indeed, tracheal aECM was not developed (not thickened) in the Muc91C-RNAi insect and it's elasticity was significantly lower than control insect. Interestingly, when the development of tracheal aECM was disrupted by Muc91C-RNAi, the total amount of trachea-residing PWN was significantly decreased, indicating that PWN-mediated tracheal hypoxia is critical to vector competence to carry more nematodes.

Thanks for your positive comments.

Although the authors have come up with fairly clear results, there are some shortcomings. A key point of this paper is the elasticity and thickness of aECM of tracheal tissue, but they just showed TEM images and mathematical constant (Young's modulus) to demonstrate them. This is ambiguous and very insufficient to interpret the importance of tracheal elasticity. The author should present more rigid and scientific results to explain this key context (e.g., at least need to do immunostaining of resilin-like mucin protein between PWN-infected and non-infected insects or other measuring methods).Nevertheless, they found a novel functional gene (Muc91C), which is partially important to tracheal development such as thickness and elasticity.

As the Reviewer suggested, we carried out immunostaining assay of Muc91C protein in PWN-infected and non-infected beetles and found PWN loading significantly enhance the expression of Muc91C protein, which formed a continuous layer beneath the layer of taenidial folds and above the apical membrane of the tracheal cell, as shown in line

290-295 and Figure 4E & F. In addition, by conducting Western blot for Muc91C in tracheal tubes with null, light and heavy PWN, we found Muc91C expression were positively correlated with PWN loading number, as shown in line 285-290 and Figure 4D. Therefore, we have provided solid evidence that the expression of *Muc91C* is responsible for thickened aECM and enhanced tracheal elasticity in beetles with heavy PWN loading.

References

Andersen SO. Studies on resilin-like gene products in insects. *Insect Biochemistry and Molecular Biology*, 2010, 40: 541-551. DOI: 10.1016/j.ibmb.2010.05.002

Berg JT, Fu Z, Breen EC, Tran H-C, Mathieu-Costello O, West JB. High lung inflation increases mRNA levels of ECM components and growth factors in lung parenchyma. 1997, 83: 120-128. DOI:10.1152/jappl.1997.83.1.120

Elvin CM, Carr AG, Huson MG, Maxwell JM, Pearson RD, Vuocolo T, Liyou NE, Wong DCC, Merritt DJ, Dixon NE. Synthesis and properties of crosslinked recombinant pro-resilin. *Nature*, 2005, 437: 999-1002. DOI:10.1038/nature04085

Glasheen BM, Robbins RM, Piette C, Beitel GJ, Page-McCaw A. A matrix metalloproteinase mediates airway remodeling in *Drosophila*. *Developmental Biology*, 2010, 344: 772-783. DOI: 10.1016/j.ydbio.2010.05.504

Greenlee KJ, Socha JJ, Eubanks HB, Pedersen P, Lee WK, Kirkton SD. Hypoxia-induced compression in the tracheal system of the tobacco hornworm caterpillar, *Manduca sexta*. *Journal of Experimental Biology*, 2013, 216: 2293-2301. DOI:10.1242/jeb.082479

Hayashi S, Kondo T. Development and function of the *Drosophila* tracheal system. *Genetics*, 2018, 209: 367-380. DOI:10.1534/genetics.117.300167

Henry JR, Harrison JF. Plastic and evolved responses of larval tracheae and mass to varying atmospheric oxygen content in *Drosophila melanogaster*. *The Journal of Experimental Biology*, 2004, 207: 3559-3567. DOI:10.1242/jeb.01189

Lyons RE, Lesieur E, Kim M, Wong DCC, Huson MG, Nairn KM, Brownlee AG, Pearson RD, Elvin CM. Design and facile production of recombinant resilin-like polypeptides: gene construction and a rapid protein purification method. *Protein Engineering, Design and Selection*, 2007, 20: 25-32. DOI:10.1093/protein/gzl050

Nairn KM, Lyons RE, Mulder RJ, Mudie ST, Cookson DJ, Lesieur E, Kim M, Lau D, Scholes FH, Elvin CM. A synthetic resilin Is largely unstructured. *Biophysical Journal*, 2008, 95: 3358-3365. DOI:10.1529/biophysj.107.119107

Öztürk-Çolak A, Moussian B, Araújo SJ. *Drosophila* chitinous aECM and its cellular interactions during tracheal development. *Developmental Dynamics*, 2016, 245: 259-267. DOI:10.1002/dvdy

Tang X, Koski T-M, Sun J. The entry of pinewood nematode is linked to programmed tracheal development of vector beetles. *bioRxiv*, 2022: 2022.2010.2029.514345. DOI:10.1101/2022.10.29.514345

Webster MR, De Vita R, Twigg JN, Socha JJ. Mechanical properties of tracheal tubes in the American cockroach (*Periplaneta americana*). *Smart Materials and Structures*, 2011, 20: 094017. DOI:10.1088/0964-1726/20/9/094017

Westneat MW, Betz O, Blob RW, Fezzaa K, Cooper WJ, Lee WK. Tracheal respiration in insects visualized with synchrotron x-ray imaging. *Science*, 2003, 299: 558-560. DOI: doi:10.1126/science.1078008

Wirtz HR, Dobbs LG. The effects of mechanical forces on lung functions. *Respiration Physiology*, 2000, 119: 1-17. DOI:10.1016/S0034-5687(99)00092-4

Zhao L, Zhang S, Wei W, Hao H, Zhang B, Butcher RA, Sun J. Chemical signals synchronize the life cycles of a plant-parasitic nematode and its vector beetle. *Current Biology*, 2013, 23: 2038-2043. DOI: 10.1016/j.cub.2013.08.041

Zhao L, Mota M, Vieira P, Butcher RA, Sun J. Interspecific communication between pinewood nematode, its insect vector, and associated microbes. *Trends in Parasitology*, 2014, 30: 299-308. DOI:10.1016/j.pt.2014.04.007

[Editors’ note: what follows is the authors’ response to the second round of review.]

The manuscript has been improved but there are some remaining issues that need to be addressed, as outlined below:1) The authors show that Muc91C localizes underneath the chitinous aECM in tracheal tubes upon infection. This is a very good result, however, the images are not clear and the manuscript would gain from higher magnification images focusing on a small section of the membrane and showing the tracheal cells in more detail.

This comment is really helpful. In this revised version, we added higher magnification images of Muc91C in tracheal system with null or heavy nematode loading, by scanning through 63x objective and setting confocal zoom factor to 4x, thus focusing on a small section of the membrane, as shown in line 319-327 and Figure 5A &B, at the bottom right panels. The images provide robust evidence that heavy nematode loading caused substantial upregulation of Muc91C beneath the layer of chitinous taenidial folds and above the apical membrane of the tracheal cell.

Reviewer #2 (Recommendations for the authors):– Reference to the elastic property of repeat A: the way the authors state that Muc91 repeat A has been shown to "regulate elasticity" is, to my understanding, wrong. First, repeat A from pro-resiling has been shown to confer elasticity, not of Muc91. Second, elasticity does not come from the repeats, but from their ability to form di-tyrosine bridges, and this has not been shown.

Repeat A in elastic pro-resilins has been proved to confer elasticity. In the revised version, we described in details the correlation between repeat A of Muc91C and elasticity in line 344-350 by adding related literature, and made revision to the presentation of causality in line 359-364. Repeat A in Muc91C of *M. alternatus* (PSSSYGAPS) contains glycine and proline residues, which can form a stretchable β-spiral structure (Ardell and Andersen, 2001; Tatham and Shewry, 2002). Besides, the tyrosines residues in repeat A of Muc91C facilitate the formation of di-tyrosine crosslinks with other repeats (Andersen, 2010; Qin et al., 2009). Thus, repeat A of Muc91C is likely to confer long-range elasticity. This has been further evidenced by an empirical study showing that peptide chains synthesized from 16 copies of similarly repeat A (AQTPSSQYGAP) in Muc91C of *A. gambiae* can from rubber-like elastic materials (Lyons et al., 2007; Nairn et al., 2008).

– My argument that Muc91 may be involved in retaining water in the cuticle and thereby contributing to elasticity was refuted. Fine with me, if this was done with a valid argument. Muc91 as a mucin may bind water and thereby contribute to elasticity, and not (only) via its repeats.

We highly appreciate this comment that contributes to a more comprehensive understanding of our results, and agree with you that mucins may bind water because of the decoration by sugars (Wagner et al., 2018) and the Muc91C which may plasticized by water coating may provide additional elasticity from the surface tension of the liquid, as proved in spider silk (Vollrath and Edmonds, 1989), in addition to repeated sequences. This possible contribution of water in elasticity of Muc91C was added in *Discussion* line 496-500.

– Young's modulus: did I understand correctly that the forces were measured along the main axis of the tracheal tubes? To me, they should be measured perpendicular to the main axis. Please describe the method more in detail. Forces may be different in different directions.

We added more details in the methods describing measurement of mechanical property of tracheal tubes in line 600-602 and provided more explanation of Young’s modulus in 603-604. The forces were measured along the main axis (the axial direction), according to the procedures of tensile testing on muscle fibres (Krysiak et al., 2018). We used Young’s module to characterize the elasticity of tracheal tubes, because this constant can specify the stiffness of a material irrespective of directionality of the pulling force used (Feynman et al., 1965).

References

Andersen SO. Studies on resilin-like gene products in insects. *Insect Biochemistry and Molecular Biology*, 2010, 40: 541-551. DOI:10.1016/j.ibmb.2010.05.002

Ardell DH, Andersen SO. Tentative identification of a resilin gene in *Drosophila melanogaster*. *Insect Biochemistry and Molecular Biology*, 2001, 31: 965-970. DOI:10.1016/S0965-1748(01)00044-3

Feynman RP, Leighton RB, Sands M. The feynman lectures on physics; vol. ii. *American Journal of Physics*, 1965, 33: 750-752.

Krysiak J, Unger A, Beckendorf L, Hamdani N, von Frieling-Salewsky M, Redfield MM, Dos Remedios CG, Sheikh F, Gergs U, Boknik P, Linke WA. Protein phosphatase 5 regulates titin phosphorylation and function at a sarcomere-associated mechanosensor complex in cardiomyocytes. *Nature Communications*, 2018, 9: 262. DOI:10.1038/s41467-017-02483-3

Lyons RE, Lesieur E, Kim M, Wong DC, Huson MG, Nairn KM, Brownlee AG, Pearson RD, Elvin CM. Design and facile production of recombinant resilin-like polypeptides: gene construction and a rapid protein purification method. *Protein Engineering, Design and Selection*, 2007, 20: 25-32. DOI:10.1093/protein/gzl050

Nairn KM, Lyons RE, Mulder RJ, Mudie ST, Cookson DJ, Lesieur E, Kim M, Lau D, Scholes FH, Elvin CM. A synthetic resilin is largely unstructured. *Biophysical Journal*, 2008, 95: 3358-3365. DOI:10.1529/biophysj.107.119107

Qin G, Lapidot S, Numata K, Hu X, Meirovitch S, Dekel M, Podoler I, Shoseyov O, Kaplan DL. Expression, cross-Linking, and characterization of recombinant chitin binding resilin. *Biomacromolecules*, 2009, 10: 3227-3234. DOI:10.1021/bm900735g

Tatham AS, Shewry PR. Comparative structures and properties of elastic proteins. *Philosophical Transactions of the Royal Society B: Biological Sciences*, 2002, 357: 229-234. DOI:10.1098/rstb.2001.1031

Vollrath F, Edmonds DT. Modulation of the mechanical properties of spider silk by coating with water. *Nature*, 1989, 340: 305-307. DOI:10.1038/340305a0

Wagner CE, Wheeler KM, Ribbeck K. Mucins and their role in shaping the functions of mucus barriers. *Annual Review of Cell and Developmental Biology*, 2018, 34: 189-215. DOI:10.1146/annurev-cellbio-100617-062818